# Echinoderms provide missing link in the evolution of PrRP/sNPF-type neuropeptide signalling

**Luis Alfonso Yañez-Guerra[1†‡], Xingxing Zhong[1†§], Ismail Moghul[1#], Thomas Butts[1¶], Cleidiane G Zampronio[2], Alexandra M Jones[2], Olivier Mirabeau[3], Maurice R Elphick[1]\***

[1]Queen Mary University of London, School of Biological and Chemical Sciences, London, United Kingdom; [2]School of Life Sciences and Proteomics Research Technology Platform, University of Warwick, Coventry, United Kingdom; [3]Cancer Genetics Unit, Institut Curie, Paris, France

**\*For correspondence:**
m.r.elphick@qmul.ac.uk

[†]These authors contributed equally to this work

**Present address:** [‡]Living Systems Institute, University of Exeter, Exeter, United Kingdom; [§]Department of Molecular Pharmacology, Diabetes Research Center, Institute of Aging, Albert Einstein College of Medicine, Bronx, United States; [#]UCL Cancer Institute, University College London, London, United Kingdom; [¶]School of Life Sciences, University of Liverpool, Liverpool, United Kingdom

**Competing interests:** The authors declare that no competing interests exist.

**Abstract** Neuropeptide signalling systems comprising peptide ligands and cognate receptors are evolutionarily ancient regulators of physiology and behaviour. However, there are challenges associated with determination of orthology between neuropeptides in different taxa. Orthologs of vertebrate neuropeptide-Y (NPY) known as neuropeptide-F (NPF) have been identified in protostome invertebrates, whilst prolactin-releasing peptide (PrRP) and short neuropeptide-F (sNPF) have been identified as paralogs of NPY/NPF in vertebrates and protostomes, respectively. Here we investigated the occurrence of NPY/NPF/PrRP/sNPF-related signalling systems in a deuterostome invertebrate phylum – the Echinodermata. Analysis of transcriptome/genome sequence data revealed loss of NPY/NPF-type signalling, but orthologs of PrRP-type neuropeptides and sNPF/PrRP-type receptors were identified in echinoderms. Furthermore, experimental studies revealed that the PrRP-type neuropeptide pQDRSKAMQAERTGQLRRLNPRF-$NH_2$ is a potent ligand for a sNPF/PrRP-type receptor in the starfish *Asterias rubens*. Our findings indicate that PrRP-type and sNPF-type signalling systems are orthologous and originated as a paralog of NPY/NPF-type signalling in Urbilateria.

## Introduction

Neuropeptides are neuronally secreted signalling molecules that regulate many physiological processes and behaviours in animals, including feeding, digestion, reproduction and social behaviour. They typically exert effects by binding to cognate G-protein coupled receptors (GPCRs) on target cells, which leads to changes in the activity of downstream effectors (e.g. ion channels, enzymes) (*Jékely et al., 2018*). Investigation of the evolution of neuropeptide signalling has revealed that many of the neuropeptide systems found in vertebrates have orthologs in invertebrate deuterostomes (urochordates, cephalochordates, hemichordates, echinoderms) and protostomes (e.g. arthropods, nematodes, molluscs, annelids, platyhelminthes). Thus, the evolutionary origin of over thirty neuropeptide signalling systems has been traced back to the common ancestor of the Bilateria (Urbilateria) (*Jékely, 2013*; *Mirabeau and Joly, 2013*; *Elphick et al., 2018*).

One of the neuropeptide systems that originated in Urbilateria is neuropeptide Y (NPY)-type signalling. NPY is a 36-residue peptide that was first isolated from the porcine hypothalamus (*Tatemoto et al., 1982*; *Tatemoto, 1982*) but which is also expressed by neurons in many other regions of the nervous system (*Adrian et al., 1983*; *Morris, 1989*) and in peripheral organs such as the gut and cardiovascular system (*Holzer et al., 2012*; *Farzi et al., 2015*). Accordingly, NPY is pleiotropic (*Pedrazzini et al., 2003*), although it is most widely known as a potent stimulant of food

intake in mammals (*Minor et al., 2009*; *Zhang et al., 2011*). NPY belongs to a family of related signalling molecules in vertebrates, including peptide YY (PYY) and pancreatic polypeptide (PP), which evolved from a common ancestral peptide by gene/genome duplication (*Larhammar et al., 1993*; *Larhammar, 1996*; *Elphick et al., 2018*). Furthermore, the sequences of NPY-type peptides are highly conserved across the vertebrates (*Larhammar et al., 1993*; *Larhammar, 1996*; *Cerdá-Reverter et al., 2000*).

A neuropeptide in vertebrates that is related to NPY/PYY/PP-type peptides is prolactin-releasing peptide (PrRP), which was first discovered as a ligand for the orphan receptor hGR3 (*Hinuma et al., 1998*). Phylogenetic analysis has revealed that PrRP-type receptors are paralogs of NPY/PYY/PP-type receptors and it has been proposed that PrRP-type signalling originated in the vertebrate lineage (*Lagerström et al., 2005*). However, more recently, orthologs of vertebrate PrRP-type receptors have been identified in invertebrate deuterostomes - the cephalochordate *Branchiostoma floridae* and the hemichordate *Saccoglossus kowalevskii* - indicating that PrRP-type signalling may have originated in a common ancestor of the deuterostomes (*Mirabeau and Joly, 2013*).

An important insight into the evolutionary history of NPY-related peptides was obtained with identification of a PP-like immunoreactive peptide in a protostome invertebrate, the platyhelminth *Moniezia expansa* (*Maule et al., 1991*). Sequencing revealed a 39-residue peptide with a similar structure to NPY, but with the C-terminal tyrosine (Y) substituted with a phenylalanine (F). Hence, this invertebrate NPY homolog was named neuropeptide F (NPF) (*Maule et al., 1991*). Subsequently, NPF-type neuropeptides have been identified in other protostome invertebrates, including other platyhelminths (*Curry et al., 1992*), molluscs (*Leung et al., 1992*; *Rajpara et al., 1992*), annelids (*Veenstra, 2011*; *Conzelmann et al., 2013*; *Bauknecht and Jékely, 2015*) and arthropods (*Brown et al., 1999*), and these peptides typically have a conserved C-terminal RPRFamide motif and range in length from 36 to 40 residues (*Fadda et al., 2019*).

Following the discovery of *M. expansa* NPF, antibodies to this peptide were generated and used to assay for related peptides in other invertebrates. Interestingly, this resulted in the discovery of two novel neuropeptides, ARGPQLRLRFamide and APSLRLRFamide, in the Colorado potato beetle *Leptinotarsa decemlineata* (*Spittaels et al., 1996*). As these peptides were isolated using antibodies to *M. expansa* NPF, they were originally referred to as NPF-related peptides. However, because they are much shorter in length than NPF, they were later renamed as short neuropeptide F (sNPF) (*Vanden Broeck, 2001*) and homologs were identified in other insects (*Schoofs et al., 2001*). Furthermore, alignment of NPY-type peptides and precursors from vertebrates with NPF-type and sNPF-type peptides and precursors from protostomes revealed that whilst NPF-type peptides are clearly orthologs of vertebrate NPY-type peptides, sNPF-type peptides and precursors exhibit too many differences to be considered orthologs of NPY/NPF-type peptides and precursors (*Nässel and Wegener, 2011*). Further evidence that chordate NPY-type and invertebrate NPF-type neuropeptides are orthologous has been provided by similarity-based clustering methods, showing that the NPY-type and NPF-type precursors form a pan-bilaterian cluster, whereas sNPF-type precursors form a separate cluster (*Jékely, 2013*). Thus, sNPF-type peptides are considered to be a family of neuropeptides that is distinct from the NPY/NPF-type family of neuropeptides.

A receptor for sNPF-type peptides was first identified in the fruit fly *Drosophila melanogaster* with the deorphanisation of the GPCR CG7395 (*Mertens et al., 2002*), which was previously annotated as a homolog of mammalian NPY-type receptors. Subsequently, sNPF receptors have been identified in other insects (*Chen and Pietrantonio, 2006*; *Garczynski et al., 2007*; *Yamanaka et al., 2008*; *Dillen et al., 2013*; *Dillen et al., 2014*; *Jiang et al., 2017*; *Ma et al., 2017*; *Christ et al., 2018*). A variety of physiological roles have been attributed to sNPF-type peptides in insects, with the most consistent being actions related to the regulation of feeding behaviour. For example, in *D. melanogaster* overexpression of sNPF increases food intake both in larvae and adults, whilst loss-of-function sNPF-mutants exhibited reduced food intake (*Lee et al., 2004*). It was initially thought that the sNPF-type neuropeptide signalling system may be unique to arthropods (*Nässel and Wegener, 2011*); however, a large-scale phylogenetic analysis of G-protein coupled neuropeptide receptors revealed that sNPF-type signalling is also present in other protostomes (*Mirabeau and Joly, 2013*). Thus, an expanded family of neuropeptide receptors in the nematode *C. elegans* that had originally been annotated as NPY/NPF-type receptors (*Cardoso et al., 2012*) were found to be orthologs of insect sNPF-receptors (*Mirabeau and Joly, 2013*). Furthermore, whilst NPY/NPF-type peptides and their receptors were identified as a bilaterian neuropeptide signalling system, it was proposed that

sNPF-type signalling may be restricted to protostomes (*Mirabeau and Joly, 2013*). Subsequently, sNPF-type peptides and a cognate receptor have been characterised in the bivalve mollusc *Crassostrea gigas*, confirming the occurrence of this signalling system in the lophotrochozoan branch of the protostomes (*Bigot et al., 2014*). Furthermore, the physiological roles of sNPF-type neuropeptides have been characterised in *C. gigas* and in other molluscs (*Hoek et al., 2005*; *Zatylny-Gaudin et al., 2010*; *Bigot et al., 2014*).

Important insights into neuropeptide evolution have been obtained recently by pharmacological characterisation of G-protein coupled neuropeptide receptors in invertebrate deuterostomes (*Kawada et al., 2010*; *Roch et al., 2014*; *Bauknecht and Jékely, 2015*; *Semmens et al., 2015*; *Tian et al., 2016*; *Yañez-Guerra et al., 2018*). However, currently little is known about the occurrence and characteristics of NPY/NPF/PrRP/sNPF-related signalling systems in invertebrate deuterostomes. Phylogenetic analysis of bilaterian G-protein coupled neuropeptide receptors has demonstrated the occurrence of NPY/NPF receptor-related proteins in ambulacrarians – the echinoderm *Strongylocentrotus purpuratus* and the hemichordate *Saccoglossus kowalevskii* (*Mirabeau and Joly, 2013*). Furthermore, the precursor of a putative NPY/NPF-type peptide was identified in *S. kowalevskii* (*Mirabeau and Joly, 2013*; *Elphick and Mirabeau, 2014*). A candidate NPY/NPF-type precursor has also been identified in the cephalochordate *Branchiostoma floridae*, but an NPY/NPF-type receptor has yet to be identified in this species (*Mirabeau and Joly, 2013*; *Elphick and Mirabeau, 2014*). A more recent finding was the discovery of a family neuropeptide precursor-type proteins in echinoderms that contain a peptide that shares sequence similarity with NPY/NPF-type peptides (*Zandawala et al., 2017*). However, it is not known if these proteins are orthologs of vertebrate NPY-type precursors and protostome NPF-type precursors. To address this issue, detailed analysis of the sequences of the echinoderm NPY/NPF-like peptides and precursors and the genes encoding these peptides/proteins is needed. Furthermore, the receptors for echinoderm NPY/NPF-like peptides need to be identified. Accordingly, here we show that NPY/NPF-type signalling has in fact been lost in echinoderms and report the discovery and pharmacological characterisation of a PrRP/sNPF-type signalling system in an echinoderm - the starfish *Asterias rubens*. These findings provide important new insights into the evolution of neuropeptide signalling in the Bilateria.

## Results

### NPY-like neuropeptides in echinoderms share sequence similarity with PrRP-type neuropeptides

The sequence of a transcript (contig 1060225; GenBank accession number MK033631.1) encoding the precursor of an NPY-like neuropeptide has been reported previously based on analysis of neural transcriptome sequence data from the starfish *A. rubens* (*Zandawala et al., 2017*). Here, a cDNA encoding this precursor was cloned and sequenced, revealing that the open reading frame encodes a 108-residue protein comprising a predicted 19-residue signal peptide, a 23-residue NPY-like peptide sequence with an N-terminal glutamine residue and a C-terminal glycine residue, followed by a putative monobasic cleavage site (*Figure 1—figure supplement 1A*). Analysis of radial nerve cord extracts using mass spectrometry (LC-MS-MS) revealed the presence of a peptide with the structure pQDRSKAMQAERTGQLRRLNPRF-NH$_2$, showing that the N-terminal glutamine and C-terminal glycine in the precursor peptide are post-translationally converted to a pyroglutamate residue and an amide group, respectively (*Figure 1—figure supplement 1B*).

Alignment of the sequences of the *A. rubens* neuropeptide and orthologs from other echinoderms with related peptides in other taxa revealed that they share sequence similarity with both PrRP-type neuropeptides (*Figure 1A*) and with NPY/NPF-type neuropeptides (*Figure 1B*). However, the echinoderm peptides comprise 22–25 residues and are similar in length to vertebrate PrRPs, which are 20–31 residues as full-length peptides and in some species can occur as N-terminally truncated peptides due the presence of a monobasic cleavage site (*Hinuma et al., 1998*; *Tachibana and Sakamoto, 2014*). This contrasts with NPY/NPF-type neuropeptides, which are longer peptides ranging in length from 36 to 40 residues (*Fadda et al., 2019*). Furthermore, by analysing sequence data from the hemichordate *S. kowalevskii* and the cephalochordate *B. floridae*, here we identified novel neuropeptides that share sequence similarity with the echinoderm

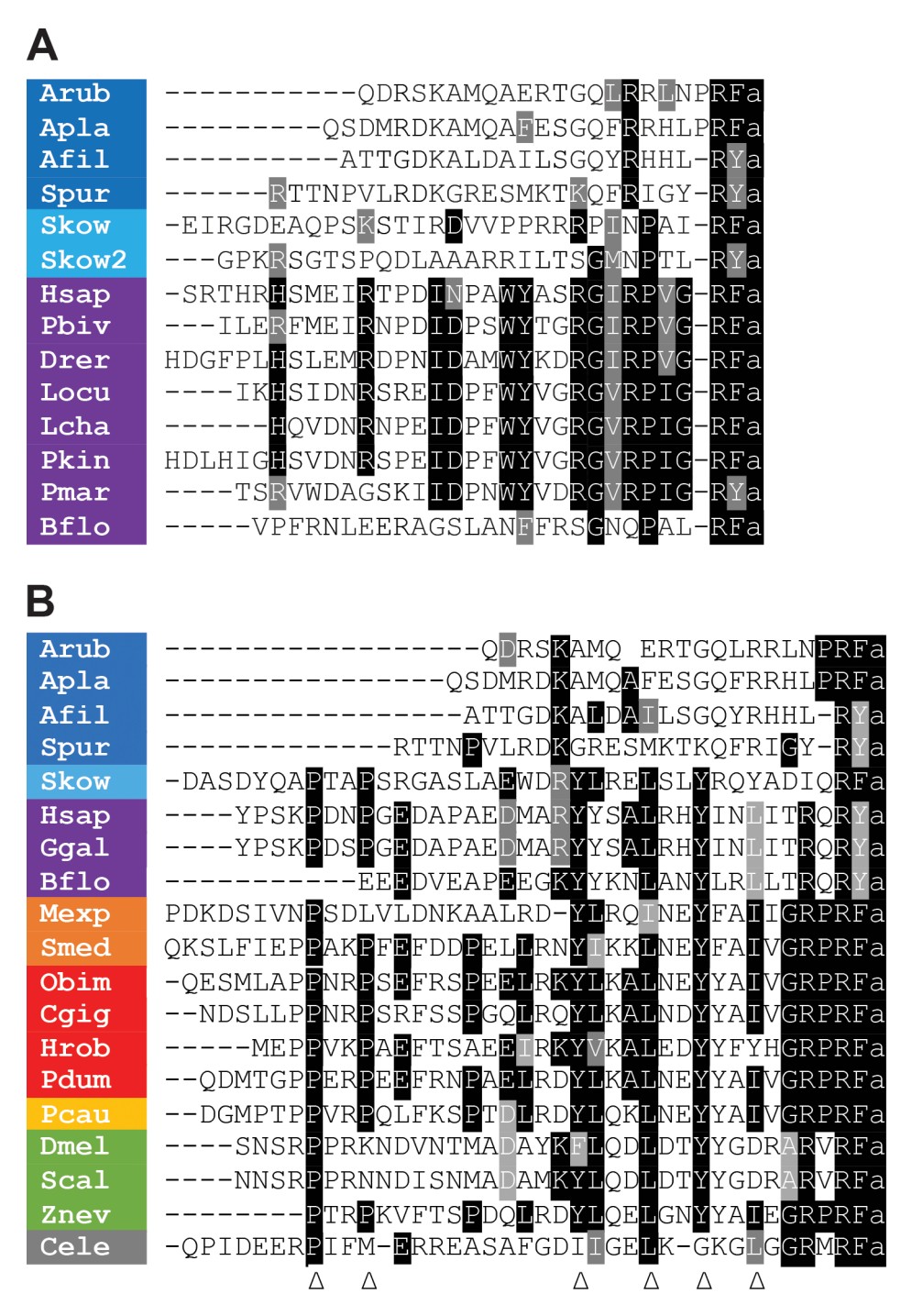

**Figure 1.** Comparison of the sequences of echinoderm NPY/NPF/PrRP-like peptides with related peptides in other taxa. (**A**) Comparison with PrRP-type neuropeptides. Conserved residues are highlighted in black (identical) or grey (conservative substitutions) (**B**) Comparison with NPY/NPF-type neuropeptides. Conserved residues are highlighted in black (identical) or grey (conservative substitutions). The arrowheads indicate residues that have been shown to be important for the three-dimensional structure of the NPY/NPF-type peptides but which are not present in the echinoderm peptides. The colour coding of phyla is as follows: dark blue (Echinodermata), light blue (Hemichordata), purple (Chordata), orange (Platyhelminthes), red (Lophotrochozoa), yellow (Priapulida), green (Arthropoda), grey (Nematoda). The full names of the species and the accession numbers of the sequences are listed in *Figure 1—source data 1*.

*Figure 1 continued on next page*

*Figure 1 continued*

The online version of this article includes the following source data and figure supplement(s) for figure 1:

**Source data 1.** Accession numbers of the precursor sequences used for the peptide alignments in *Figure 1*.

**Figure supplement 1.** Sequencing of the *A. rubens* NPY/NPF/PrRP-like precursor cDNA and determination of the structure of the neuropeptide derived from the precursor.

**Figure supplement 1—source data 1.** Fragmentation table of the mass spectrum for the *A. rubens* neuropeptide ArPrRP (QDRSKAMQAERTGQLRRLNPRF) shown in *Figure 1—figure supplement 1B*.

neuropeptides and with vertebrate PrRPs (*Figure 1A*). Thus, sequence alignment reveals that, in addition to a shared characteristic of a C-terminal RFamide or a RYamide (Y and F being conservative substitutions), there are other residues in the echinoderm peptides that are identical or structurally similar to equivalently positioned residues in chordate PrRPs (*Figure 1A*). Contrastingly, the echinoderm peptides lack two proline (P) residues that are a conserved feature of the N-terminal region of many NPY/NPF-type peptides, with the exception of some peptides that have only one of these proline residues and a peptide in the cephalochordate *Branchiostoma floridae* that has neither (*Figure 1B*). Furthermore, there are four other residues that are highly conserved in bilaterian NPY/NPF-type peptides - tyrosine (Y), leucine (L), tyrosine (Y), and isoleucine (I) residues, which are marked with arrowheads in *Figure 1B*. These residues have been shown to be important for the formation of the three-dimensional structure in vertebrate NPY-type peptides (*Blundell et al., 1981*; *Glover et al., 1983*; *Glover et al., 1984*; *Allen et al., 1987*), so these residues may likewise be important for NPF receptor activation and bioactivity. Importantly, none of these residues are present in the echinoderm peptides.

It is noteworthy, however, that all but one of the aforementioned six conserved residues in NPY/NPF-type peptides are present in a peptide from a species belonging to a sister phylum of the echinoderms – the hemichordate *Saccoglossus kowalevskii* (*Figure 1B*; *Mirabeau and Joly, 2013*; *Elphick and Mirabeau, 2014*). Collectively these findings indicate that the echinoderm neuropeptides originally described as NPY-type peptides (*Zandawala et al., 2017*) are not orthologs of NPY/NPF-type peptides but are orthologs of chordate PrRP-type peptides. Therefore, henceforth we will refer to the *A. rubens* neuropeptide pQDRSKAMQAERTGQLRRLNPRF-NH$_2$ as ArPrRP and we will refer to orthologs in other echinoderms equivalently.

## Echinoderm PrRP-like peptide genes have the same exon-intron structure as chordate PrRP genes

To investigate further the proposition that ArPrRP and other echinoderm PrRP-like neuropeptides are orthologs of chordate PrRPs, we compared the exon-intron structure of genes encoding these peptides (*Figure 2*). This revealed that a common characteristic is the presence of an intron that interrupts the coding sequence at a position corresponding to the N-terminal or central region of the echinoderm PrRP-like peptides and vertebrate PrRPs. Furthermore, in echinoderm PrRP-like peptide genes and vertebrate PrRP genes the intron interrupts the coding sequence in the same frame, at a position between the first and second nucleotide of the interrupted codon (a phase one intron), which is denoted by +1 in *Figure 2*. Genes encoding novel precursors of PrRP-like peptides in *S. kowalevskii* and *B. floridae* also have a phase one intron. Furthermore, in the *B. floridae* gene and in one of the *S. kowalevskii* genes (Skow 2) the intron is located in the region of the gene encoding the N-terminal part of the neuropeptide, whereas in the other *S. kowalevskii* gene (Skow1) the intron is located in a region encoding the C-terminal part of the neuropeptide. The presence of a conserved intron in the same frame in echinoderm PrRP-like peptide genes, the two *S. kowalevskii* PrRP-like peptide genes and chordate PrRP-type genes supports our hypothesis that the echinoderm and hemichordate PrRP-like peptides are orthologs of chordate PrRP-type neuropeptides.

By way of comparison, echinoderm PrRP-like peptide genes have a different exon-intron structure to NPY/NPF genes. Previous studies have reported that a conserved feature of NPY/NPF genes is an intron that interrupts the coding sequence for NPY/NPF-type peptides, with the intron located between the second and third nucleotide of the codon for the arginine residue of the C-terminal RF or RY dipeptide (*Mair et al., 2000*). Here we show this conserved feature in NPY/NPF genes in

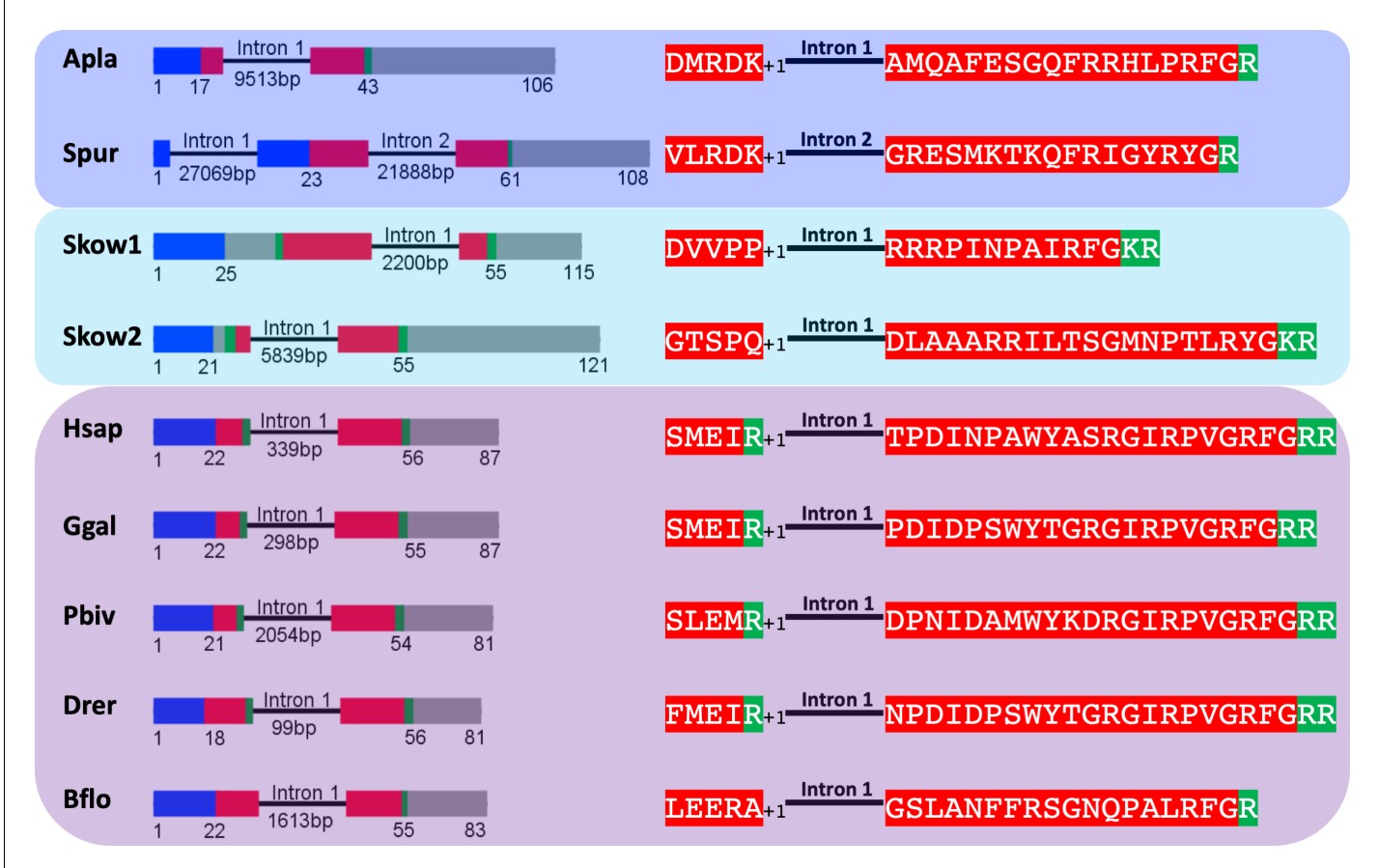

**Figure 2.** Comparison of exon/intron structure of genes encoding precursors of PrRP-like peptides in echinoderms and hemichordates with precursors of PrRP-type peptides in chordates. Schematic representations of the gene structures are shown, with protein-coding exons shown as rectangles and introns shown as lines (with intron length stated underneath). The protein-coding exons are colour-coded to show regions that encode the N-terminal signal peptide (blue), the neuropeptide (red), monobasic or dibasic cleavage sites (green) and other regions of the precursor protein (grey). Note that a common characteristic is that an intron interrupts the coding sequence in the N-terminal or central region of the neuropeptide, with the intron consistently located between the first and second nucleotides (phase one intron represented by +1) of the codon for the amino acid shown after intron. Taxa are highlighted in phylum-specific colours: dark blue (Echinodermata), light blue (Hemichordata), purple (Chordata). The full names of the species and the accession numbers of the sequences are listed in *Figure 2—source data 1*.

The online version of this article includes the following source data and figure supplement(s) for figure 2:

**Source data 1.** Accession numbers of the sequences used for the gene structure analysis in *Figure 2* and *Figure 2—figure supplement 1*.

**Figure supplement 1.** Comparison of the exon/intron structure of genes encoding echinoderm precursors of PrRP-like peptides and genes encoding NPY/NPF-type precursors in other taxa.

species from several animal phyla, including a hemichordate (sister phylum to the echinoderms), chordates, molluscs, an annelid, a priapulid, an arthropod and a nematode (*Figure 2—figure supplement 1*). In echinoderm PrRP-like peptide genes, the exon encoding the neuropeptide is likewise interrupted by an intron but it is located in a different position to the intron that interrupts the coding sequence for NPY/NPF-type peptides. Thus, it does not interrupt the codon for the arginine of the C-terminal RF or RY motif, but instead it is located between the first and second nucleotide of the codon for a residue located in the N-terminal or central regions of echinoderm PrRP-like peptides (*Figure 2—figure supplement 1*). Another difference is that typically in NPY/NPF genes there is another intron that interrupts the coding sequence in the C-terminal region of the precursor protein, whereas in the echinoderm PrRP-like peptide precursor genes the coding sequence for the C-terminal region of the precursor protein is not interrupted by an intron (*Figure 2—figure supplement 1*). Collectively, these findings provide further evidence that echinoderm PrRP-like peptides are not orthologs of NPY/NPF-type neuropeptides.

## Discovery of orthologs of sNPF/PrRP-type receptors in *A. rubens* and other echinoderms

Having obtained evidence that echinoderm NPY/PrRP-like peptides are not orthologs of NPY/NPF-type neuropeptides but are orthologs of PrRP-type peptides, we then investigated the occurrence in *A. rubens* and other echinoderms of proteins related to GPCRs that mediate effects of NPY/NPF-type peptides, PrRP-type peptides and sNPF-type peptides in other bilaterians. Using receptor sequences of *H. sapiens* NPY-type, *D. melanogaster* NPF-type, *H. sapiens* PrRP-type and *D. melanogaster* sNPF-type receptors as queries for similarity-based analysis of *A. rubens* neural

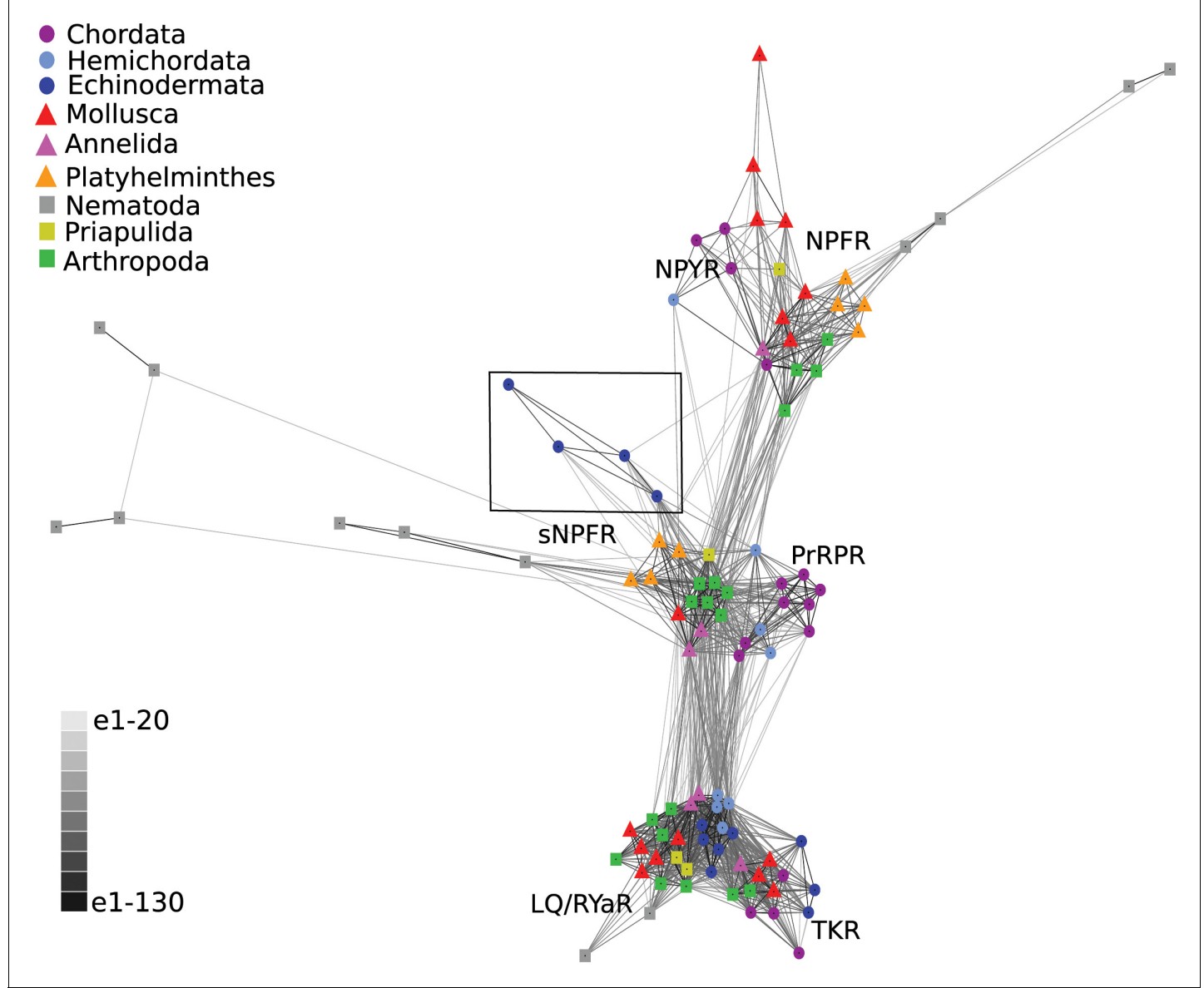

**Figure 3.** BLOSUM62 cluster map of NPY/NPF/PrPR/sNPF-type receptors and closely related tachykinin-type receptors (TKR) and luqin/RYamide-type receptors (LQ/RYaR). Nodes are labelled with phylum-specific colours, as shown in the key, and connections represent BLAST relationships with a *P* value > 1e-65. Note that the echinoderm receptors (boxed) have more connections with PrRP/sNPF-type receptors than with NPY/NPF-type receptors. The sequences of the receptors included in this figure are listed in *Figure 3—source data 1*.

The online version of this article includes the following source data and figure supplement(s) for figure 3:

**Source data 1.** Accession numbers of the receptor sequences used for the CLANS analysis in *Figure 3*.

**Figure supplement 1.** *Asterias rubens* sNPF/PrRP-type receptor (Ar-sNPF/PrRPR) transcript.

transcriptome sequence data, a transcript (contig 1120879) encoding a 386-residue protein was identified as the best hit (*Figure 3—figure supplement 1*). Furthermore, homologs of the *A. rubens* protein encoded by contig 1120879 were also identified in other echinoderms for which genome sequences have been obtained, including the starfish *A. planci*, the sea urchin *S. purpuratus* and the sea cucumber *A. japonicus*, and importantly no other NPY/NPF/PrRP/sNPF-type receptors were identified in these species. To investigate relationships of the novel echinoderm receptors with other bilaterian neuropeptide receptors, we generated a sequence database including bilaterian NPY/NPF/PrRP/sNPF-type receptors and other closely related receptors (tachykinin-type, luqin-type receptors) as outgroups. These receptor sequences were then analysed using two different methodologies.

Firstly, we performed a cluster-based analysis of the receptor sequences using CLANS (*Figure 3*). This analysis revealed three main clusters: 1. a cluster comprising the outgroup receptors (tachykinin/luqin), 2. a cluster comprising NPY/NPF-type receptors and 3. a cluster comprising sNPF-type receptors and PrRP-type receptors. Interestingly, the echinoderm receptors showed stronger connections with the sNPF/PrRP receptor cluster (*Figure 3*, black square) than with the NPY/NPF receptor cluster. These findings indicate that sNPF-type receptors and PrRP-type receptors are orthologous, as has been proposed previously based on cluster-based analysis of receptor sequences (*Jékely, 2013*). Furthermore, these findings indicate that NPY/NPF/PrRP/sNPF-type receptors in echinoderms are not orthologs of NPY/NPF-type receptors but are orthologs of sNPF/PrRP-type receptors. However, it is noteworthy that the lines linking the echinoderm receptors and nematode sNPF-type receptors with other sNPF/PrRP-type receptors in CLANS are quite long (*Figure 3*), which is indicative of sequence divergence.

Secondly, we performed a phylogenetic analysis of the receptor sequences using the maximum likelihood method. For this analysis, in addition to bilaterian NPY/NPF-type receptors, deuterostome PrRP-type receptors and protostome sNPF-type receptors, we included tachykinin-type, luqin-type and GPR83-type receptors as outgroups. This revealed that the echinoderm receptors are positioned within a branch of the phylogenetic tree that comprises NPY/NPF-type, PrRP-type and sNPF-type receptors, with the other receptor types included in the analysis occupying an outgroup position (*Figure 4*). More specifically, the echinoderm receptors are positioned in a clade comprising sNPF-type receptors, with bootstrap support of >90%, indicating that the echinoderm receptors are orthologs of protostome sNPF-type receptors. However, it is noteworthy that sNPF-type receptors and PrRP-type receptors do not form a monophyletic clade as would be expected for orthologous receptors. This may be a consequence of sequence divergence in the echinoderm and nematode sNPF/PrRP-type receptors that is reflected in the long branches leading to these receptors.

Because the phylogenetic analysis revealed that the echinoderm receptors are positioned in a clade comprising protostome sNPF-type receptors (*Figure 4*), we also compared the sequences of echinoderm PrRP-type peptides and protostome sNPF-type peptides (*Figure 4—figure supplement 1*) and the structures of the genes encoding these neuropeptides (*Figure 4—figure supplement 2*). This revealed that sequence identity is restricted to a few residues in the C-terminal regions of the peptides and, furthermore, the echinoderm PrRP-type peptides are much longer than protostome sNPF-type peptides (*Figure 4—figure supplement 1*). This contrasts with the much higher levels of sequence similarity shared between echinoderm PrRP-type neuropeptides and chordate PrRP-type neuropeptides, as shown in *Figure 1A*. Another difference is that protostome sNPF-type neuropeptide precursors typically give rise to multiple sNPF-type peptides, whereas echinoderm PrRP-type precursors are similar to chordate PrRP-type precursors in containing a single PrRP-type neuropeptide that is located adjacent to the signal peptide (*Figure 4—figure supplement 1*). Accordingly, comparison of the exon/intron structure of the genes encoding PrRP-type precursors in echinoderms and sNPF-type precursors in protostomes also revealed limited similarity (*Figure 4—figure supplement 2*).

Collectively, our analysis of sequence data indicates that NPY/NPF/PrRP/sNPF-type receptors in echinoderms are not orthologs of NPY/NPF-type receptors but are orthologs of sNPF/PrRP-type receptors. Therefore, henceforth we refer to these echinoderm receptors as sNPF/PrRP-type receptors and specifically refer to the sNPF/PrRP-type receptor in the starfish *A. rubens* as Ar-sNPF/PrRPR. Furthermore, having identified Ar-sNPF/PrRPR we proceeded to investigate if ArPrRP acts as a ligand for this receptor.

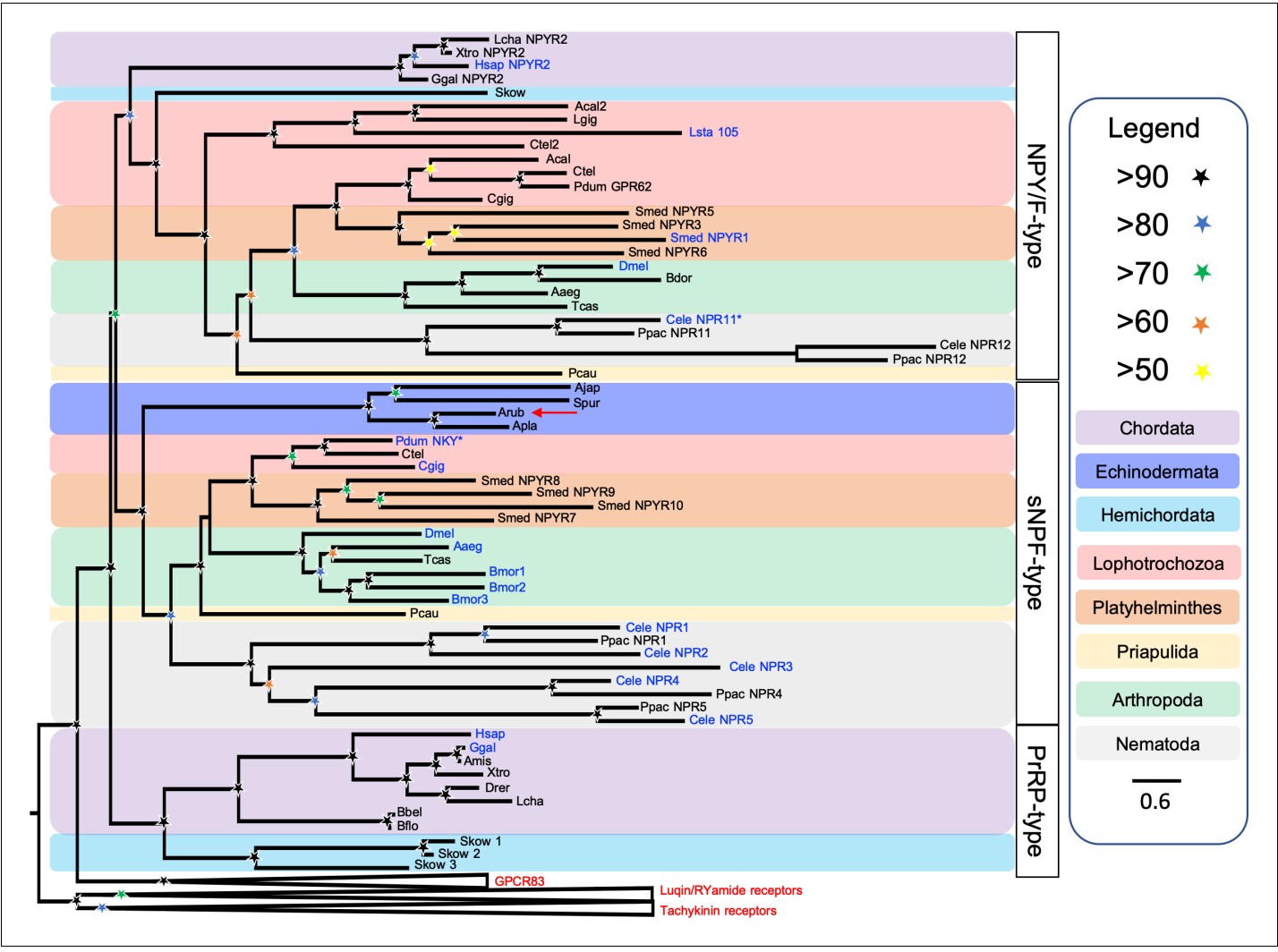

**Figure 4.** Phylogenetic tree showing that a candidate receptor for the *A. rubens* neuropeptide ArPrRP is an ortholog of protostome sNPF-type receptors. The tree includes NPY/NPF-type receptors, chordate PrRP-type receptors and protostome sNPF-type receptors, with GPR83-type, luqin-type, and tachykinin-type receptors as outgroups to root the tree. Interestingly, the candidate receptor for the *A. rubens* neuropeptide ArPrRP (red arrow) and orthologs from other echinoderms are positioned in a clade comprising protostome sNPF-type receptors, whereas candidate receptors for PrRP-type peptides in the hemichordate *S. kowalevskii* are positioned in a clade containing chordate PrRP-type receptors. Note that NPY/NPF-type receptors form a distinct clade that includes an NPY/NPF-type receptor from the hemichordate *S. kowalevskii*, but no echinoderm receptors are present in this clade. The tree was generated in W-IQ-tree 1.0 using the Maximum likelihood method. The stars represent bootstrap support (1000 replicates, see legend) and the coloured backgrounds represent different taxonomic groups, as shown in the key. The names with text in blue represent the receptors for which ligands have been experimentally confirmed. The asterisks highlight receptors where the reported ligand is atypical when compared with ligands for receptors in the same clade. Species names are as follows: Aaeg (*Aedes aegypti*), Acal (*Aplysia californica*), Ajap (*Apostichopus japonicus*), Amis (*Alligator mississippiensis*), Apla (*Acanthaster planci*), Arub (*Asterias rubens*), Bbel (*Branchiostoma belcheri*), Bdor (*Bactrocera dorsalis*), Bflo (*Branchiostoma floridae*), Bmor (*Bombyx mori*), Cele (*Caenorhabditis elegans*), Cgig (*Crassostrea gigas*), Ctel (*Capitella teleta*), Dmel (*Drosophila melanogaster*), Drer (*Danio rerio*), Ggal (*Gallus gallus*), Hsap (*Homo sapiens*), Lcha (*Latimeria chalumnae*), Lgig (*Lottia gigantea*), Lsta (*Lymnaea stagnalis*), Pcau (*Priapulus caudatus*), Pdum (*Platynereis dumerilii*), Ppac (*Pristionchus pacificus*), Skow (*Saccoglossus kowalevskii*), Smed (*Schmidtea mediterranea*), Spur (*Strongylocentrotus purpuratus*), Tcas (*Tribolium castaneum*), Xtro (*Xenopus tropicalis*). The accession numbers of the sequences used for this phylogenetic tree are listed in *Figure 4—source data 1*.

The online version of this article includes the following source data and figure supplement(s) for figure 4:

**Source data 1.** Accession numbers of the receptor sequences used for the phylogenetic analysis shown in *Figure 4*.

**Figure supplement 1.** Comparison of the sequences of ArPrRP and orthologs from other echinoderms with protostome sNPF-type peptides.

**Figure supplement 1—source data 1.** Accession numbers of the precursor sequences used for the peptide alignments in *Figure 4—figure supplement 1*.

**Figure supplement 2.** Comparison of the exon/intron structure of genes encoding echinoderm orthologs of the ArPrRP precursor and genes encoding protostome sNPF-type precursors.

*Figure 4 continued on next page*

*Figure 4 continued*

**Figure supplement 2—source data 1.** Accession numbers of the precursor sequences used for the gene structure analysis in *Figure 4—figure supplement 2*.

## Pharmacological characterisation of Ar-sNPF/PrRPR

A cDNA encoding Ar-sNPF/PrRPR was cloned and sequenced (*Figure 3—figure supplement 1*) and its sequence has been deposited in GenBank under accession number MH807444.1. Analysis of the sequence of Ar-sNPF/PrRPR using Protter revealed seven predicted transmembrane domains, as expected for a GPCR (*Figure 5—figure supplement 1*). The cloned receptor was then co-expressed with G$\alpha$16 in CHO-K1 cells expressing apoaequorin to produce the cell system CHO-Ar-sNPF/PrRPR. Synthetic ArPrRP (pQDRSKAMQAERTGQLRRLNPRF-NH$_2$) was tested as a candidate ligand for Ar-sNPF/PrRPR at concentrations ranging from $10^{-14}$ M to $10^{-5}$ M, comparing with cells incubated in assay media without the addition of the peptide. This revealed that ArPrRP at a concentration of $10^{-5}$ M triggers luminescence responses (defined as 100%) in CHO-Ar-sNPF/PrRPR cells that were approximately five times the background luminescence detected with the assay media used to dissolve the peptide (*Figure 5A*), demonstrating that ArPrRP acts as a ligand for the receptor. Furthermore, ArPrRP induced dose-dependent luminescence in CHO-Ar-sNPF/PrRPR cells with a half-maximal response concentration (EC$_{50}$) of $1.5 \times 10^{-10}$ M (*Figure 5B*). Importantly, no response to ArPrRP was observed in CHO-K1 cells transfected with the vector alone, demonstrating that the signal observed in CHO-Ar-sNPF/PrRPR cells exposed to ArPrRP can be attributed to activation of the transfected receptor (*Figure 5—figure supplement 2*). Because ArPrRP contains a potential dibasic cleavage site (see underlined arginine residues in its sequence: pQDRSKAMQAERTGQL<u>RR</u>LNPRF-NH$_2$), we hypothesised that the C-terminal pentapeptide of ArPrRP (LNPRFamide) may also be generated from ArPrRPP in vivo. Therefore, we also tested synthetic LNPRFamide as a candidate ligand for Ar-sNPF/PrRPR. However, this peptide did not induce luminescence responses in CHO-Ar-sNPF/PrRPR cells (*Figure 5B*). Therefore, we conclude that the 22-residue amidated peptide ArPrRP is the natural ligand for Ar-sNPF/PrRPR in *A. rubens.* The *A. rubens* luqin-type neuropeptide ArLQ also did not induce luminescence responses in CHO-Ar-sNPF/PrRPR cells, demonstrating the selectivity of Ar-sNPF/PrRPR for ArPrRP as a ligand (*Figure 5B*).

## Discussion

### Loss of NPY/NPF-type neuropeptide signalling in echinoderms

The discovery of an NPY-like neuropeptide, named NPF, in a platyhelminth provided the first definitive molecular evidence that NPY-type neuropeptides originated in a common ancestor of the Bilateria (*Maule et al., 1991*). Subsequently, analysis of transcriptomic/genomic sequence data has enabled identification of NPY/NPF-type neuropeptides and their cognate receptors in a variety of invertebrate taxa, revealing a high level of conservation of this signalling system in bilaterian phyla (*Zatylny-Gaudin and Favrel, 2014*; *Fadda et al., 2019*). Here we report the first detailed analysis NPY/NPF-related signalling systems in echinoderms - invertebrate deuterostomes that have provided key insights into the evolution of other neuropeptide signalling systems (*Semmens et al., 2015*; *Tian et al., 2016*; *Elphick et al., 2018*; *Yañez-Guerra et al., 2018*).

Recently, we reported the discovery of echinoderm proteins comprising putative neuropeptides that share sequence similarity with NPY/NPF-type peptides (*Zandawala et al., 2017*). However, here our detailed analysis of the sequences of these peptides and the genes encoding them has revealed that they are not orthologs of the NPY/NPF-type neuropeptides. Consistent with this finding, orthologs of NPY/NPF-type receptors were also not found in echinoderms. Therefore, we conclude that NPY/NPF-type neuropeptide signalling has been lost in the phylum Echinodermata (*Figure 6*). This is a noteworthy because, to the best of our knowledge, the only other taxon in which loss of NPY/NPF-type signalling has been reported are the urochordates, a sub-phylum of the phylum Chordata (*Mirabeau and Joly, 2013*; *Figure 6*). The evolutionary and functional significance of loss of NPY/NPF-type signalling in echinoderms and urochordates is

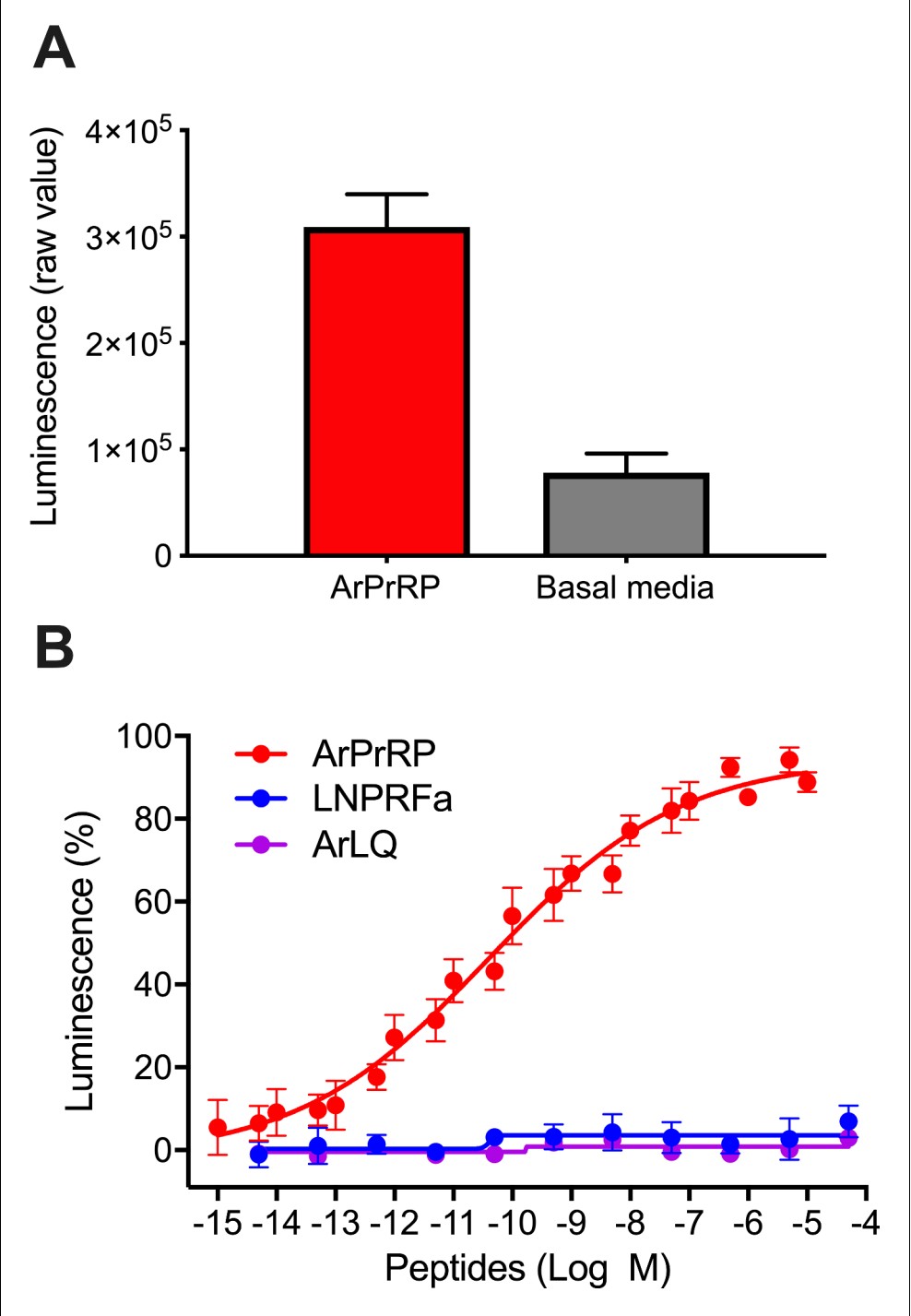

**Figure 5.** *A. rubens* PrRP-like peptide ArPrRP acts as a ligand for the *A. rubens* sNPF/PrRP-type receptor Ar-sNPF/PrRPR. (**A**) The A. rubens PrRP-like peptide ArPrRP ($10^{-5}$ M; red bar) triggers luminescence in CHO-K1 cells expressing the *A. rubens* PrRP/sNPF-type receptor Ar-sNPF/PrRPR, the promiscuous G-protein $G_{\alpha 16}$ and the calcium-sensitive luminescent GFP-apoaequorin fusion protein G5A. For comparison, the background luminescence of cells that were not exposed to ArPrRP is shown (basal media; grey bar). Mean values (± S.E.M) were determined from three independent experiments performed in triplicate (**B**). Graph showing the selectivity of ArPrRP as a ligand for Ar-sNPF/PrRPR. ArPrRP causes dose-dependent luminescence in CHO-K1 cells expressing Ar-sNPF/PrRPR, with an $EC_{50}$ of 0.15 nM. Ar-sNPF/PrRPR is not activated by a C-terminal pentapeptide fragment of ArPrRP (LNPRFamide) or by the *A. rubens* luqin-type peptide ArLQ. Each point represents mean values (± S.E.

*Figure 5 continued on next page*

*Figure 5 continued*

M) from at least three independent experiments done in triplicate. The raw data for the experiments shown in *Figure 5* and in *Figure 5—figure supplement 2* can be found in *Figure 5—source data 1*.

The online version of this article includes the following source data and figure supplement(s) for figure 5:

**Source data 1.** Data for the graphs shown in *Figure 5* and *Figure 5—figure supplement 2*.
**Figure supplement 1.** Predicted topology of Ar-sNPF/PrRPR.
**Figure supplement 2.** ArPrRP does not trigger luminescence in CHO-K1 cells transfected with an empty pcDNA 3.1(+) vector.

unknown. However, insights into this issue may emerge from functional characterisation of NPY/NPF-type signalling in other invertebrates.

The nematode *C. elegans* is a powerful model system for functional characterisation of neuropeptide signalling systems (*Frooninckx et al., 2012*). However, NPY/NPF-type signalling has thus far only been partially characterised in this species. Here, our phylogenetic analysis (*Figure 4*) indicates that there are two *C. elegans* receptors that are orthologs of NPY/NPF-type receptors: NPR-12, which is an orphan receptor, and NPR-11, which has been shown to be activated by the peptide MDANAFRMSFamide (*Chalasani et al., 2010*). However, this peptide shares little sequence similarity with NPY/NPF-type peptides from other bilaterians. Furthermore, receptor assays only showed activation at peptide concentrations of 10 and 30 μM (*Chalasani et al., 2010*), which are high when compared to other NPY/NPF-type receptors that are typically activated by ligands in the nanomolar range (*Bard et al., 1995*; *Lundell et al., 1997*; *Garczynski et al., 2002*; *Saberi et al., 2016*). Recently, based on similarity-based sequence alignments, it has been suggested that the mature peptide derived from the *C. elegans* protein FLP-27 may be an ortholog of NPY/NPF-type peptides (*Fadda et al., 2019*). Here, our analysis of the structure of the gene encoding the FLP-27 precursor has revealed that it has the characteristic structure of NPY/NPF-type genes, with an intron interrupting the codon for the C-terminal arginine of the NPF-type peptide sequence (*Figure 2—figure supplement 1*). Thus, based on our analysis of *C. elegans* sequence data, we conclude that the NPY/NPF-type peptide derived from the FLP-27 precursor protein is likely to act as a ligand for the NPR-11 and/or NPR-12 receptors. This finding provides a basis for functional characterisation of NPY/NPF-type signalling in *C. elegans*.

## Discovery of a PrRP/sNPF-type neuropeptide signalling system in echinoderms

If the echinoderm NPY-like peptides are not orthologs of NPY/NPF-type neuropeptides, then what are they? Here we show that these peptides share sequence similarity with vertebrate PrRP-type neuropeptides (*Figure 1A*). Furthermore, analysis of the structure of the genes encoding the echinoderm neuropeptides revealed that the coding sequence for the neuropeptides is interrupted by an intron in the phase one frame, a feature that is also a characteristic of genes encoding vertebrate PrRP-type neuropeptides (*Figure 2*). These findings indicate that the echinoderm neuropeptides are orthologs of vertebrate PrRP-type neuropeptides. To further address this issue we analysed echinoderm genome/transcriptome sequence data to identify candidate cognate receptors for the echinoderm PrRP-like peptides. A cluster-based analysis of receptor sequence data using CLANS revealed the presence in echinoderms of receptor proteins that show strong connections with a receptor cluster comprising vertebrate PrRP-type receptors and protostome sNPF-type receptors (*Figure 3*). Accordingly, a previous cluster-based analysis of receptor sequence data has reported that vertebrate PrRP-type receptors cluster with protostome sNPF-type receptors, indicating that these receptors may be orthologous (*Jékely, 2013*). A novelty of our analysis is the inclusion of several echinoderm receptor sequences. It is noteworthy, however, that whilst strong connections between the echinoderm receptors and PrRP/sNPF-type receptors in other taxa can be seen using CLANS, the lines linking to the echinoderm receptors are quite long (*Figure 3*). This suggests that the echinoderm receptors are orthologs of PrRP/sNPF-type receptors but have undergone sequence divergence. Interestingly, a group of sNPF-type receptors in the nematode *C. elegans* appears to be similarly divergent with respect to other sNPF/PrRP-type receptors (*Figure 3*).

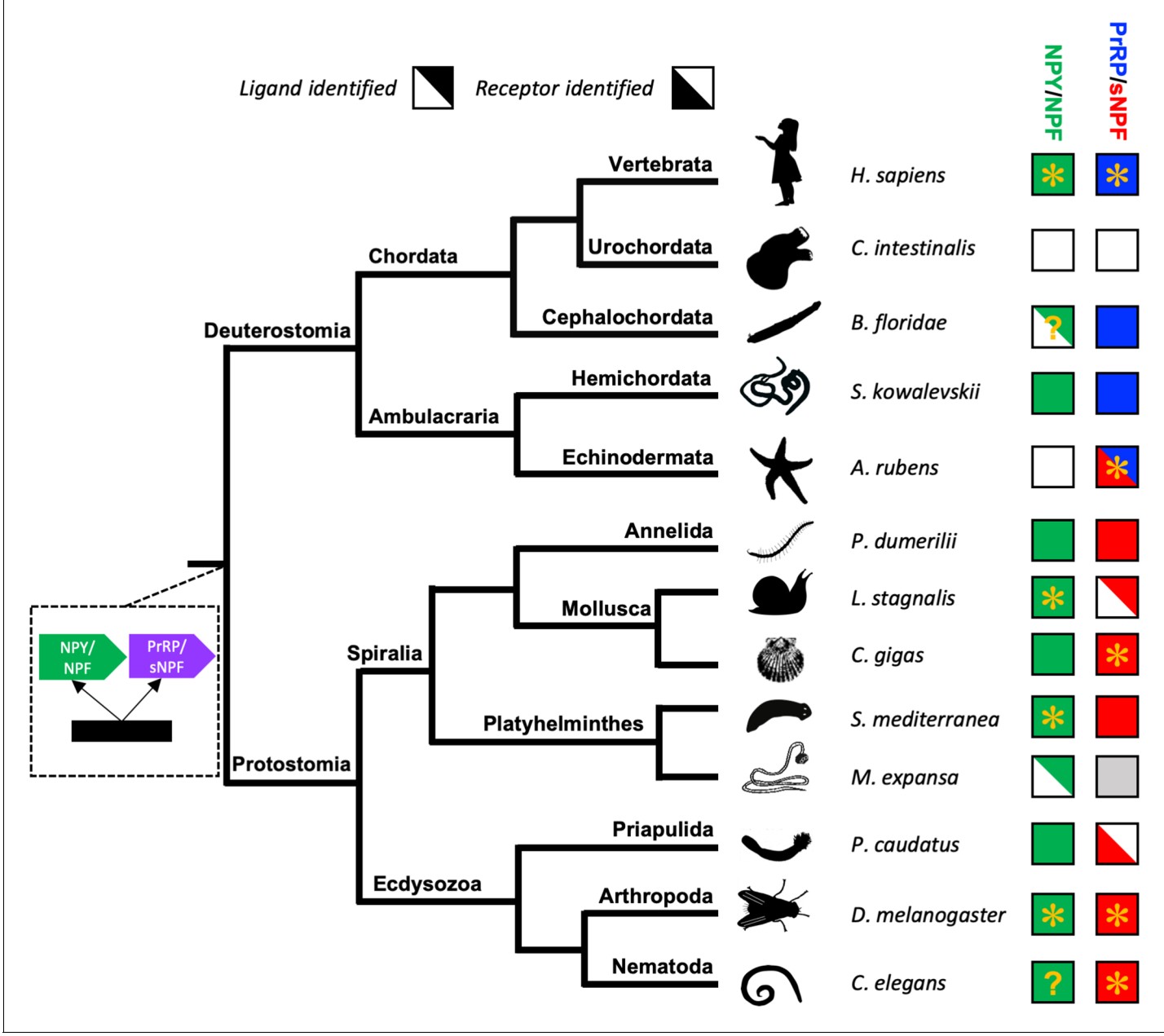

**Figure 6.** Phylogenetic diagram showing the occurrence of NPY/NPF-type, sNPF-type and PrRP-type neuropeptide signalling in the Bilateria. The tree shows the phylogenetic relationships of selected bilaterian phyla. A gene duplication event giving rise to the paralogous NPY/NPF-type (green) and PrRP/sNPF (purple) signalling systems is shown at a position in the tree corresponding to the common ancestor of the Bilateria. Phyla in which NPY/NPF-type peptides/precursors and NPY/NPF-type receptors have been identified are labelled with green-filled squares. Phyla in which PrRP-type peptides/precursors and PrRP-type receptors have been identified are labelled with blue-filled squares. Phyla in which sNPF-type peptides/precursors and sNPF-type receptors have been identified are labelled with red-filled squares. The inclusion of an asterisk in filled squares indicates that activation of a receptor by a peptide ligand has been demonstrated experimentally. Note that in the starfish *Asterias rubens* (this study) a PrRP-type peptide (blue triangle) is the ligand for receptor that has been found to be an ortholog sNPF/PrRP-type receptors (*Figure 3*) or an ortholog of sNPF-type receptors (*Figure 4*); hence this receptor is represented here as a red triangle. Note also the mutually exclusive patterns in the phylogenetic distribution of sNPF-type signalling and PrRP-type signalling, with the former found in protostomes and the latter found in vertebrates, cephalochordates and hemichordates, which is supportive of the hypothesis that these signalling systems are orthologous. Our discovery of a PrRP/sNPF-type signalling system in echinoderms provides a missing link in the evolution of this neuropeptide signalling system. NPY/NPF-type signalling occurs in most phyla, but it has been lost in echinoderms and urochordates. The inclusion of a question mark for the putative NPY/NPF-type peptide identified in the cephalochordate *B. floridae* (***Mirabeau and Joly, 2013***; ***Elphick and Mirabeau, 2014***) signifies that it is atypical of NPY/NPF-type peptides, which may explain why NPY/NPF-type receptors have yet to be identified in cephalochordates. The inclusion of a question mark in the *C. elegans* green square

*Figure 6 continued on next page*

*Figure 6 continued*

indicates that the peptide identified as a ligand for the *C. elegans* NPY/NPF-type receptor (**Chalasani et al., 2010**) does not have the typical features of an NPY/NPF-type peptide. The grey square for sNPF in *M. expansa*, for which only transcriptome sequence data are available, indicates that sNPF-type peptides and sNPF-type receptor(s) are likely to be present in this species because sNPF-type peptides and sNPF-type receptors have been identified in another platyhelminth species, *S. mediterranea*, for which a genome sequence is available. Species names are as follows: *H. sapiens* (*Homo sapiens*), *C. intestinalis* (*Ciona intestinalis*), *B. floridae* (*Branchiostoma floridae*), *S. kowalevskii* (*Saccoglossus kowalevskii*), *A. rubens* (*Asterias rubens*), *P. dumerilii* (*Platynereis dumerilii*), *L. stagnalis* (*Lymnaea stagnalis*), *M. expansa* (*Moniezia expansa*), *S. mediterranea* (*Schmidtea mediterranea*), *C. gigas* (*Crassostrea gigas*), *D. melanogaster* (*Drosophila melanogaster*), *C. elegans* (*Caenorhabditis elegans*). Silhouettes of representative animals from each phylum are from www.openclipart.com and they are free from copyright.

To further investigate the relationship of the echinoderm receptors with sNPF/PrRP-type receptors, we performed a phylogenetic analysis of sequence data using the maximum likelihood method (*Figure 4*). In this analysis, the echinoderm receptors are positioned in a clade comprising protostome sNPF-type receptors. However, sNPF-type receptors and PrRP-type receptors do not form a monophyletic clade in the tree. Interestingly, this finding has been reported previously as part of a wider analysis of neuropeptide receptor relationships in the Bilateria (**Mirabeau and Joly, 2013**). Thus, there is inconsistency in the findings from cluster-based analysis (CLANS) (**Jékely, 2013**; *Figure 3*) and phylogenetic tree-based analysis (**Mirabeau and Joly, 2013**; *Figure 4*) of receptor relationships. One possible explanation for this inconsistency would be that gene duplication in a common ancestor of the Bilateria gave rise to two sNPF/PrRP-type signalling systems, which were then differentially lost/retained in bilaterian lineages, but in such a scenario gene loss in several lineages would have to be invoked. Alternatively, the inconsistency may, at least in part, be a consequence of sequence divergence in echinoderm and nematode sNPF/PrRP-type receptors with respect sNPF/PrRP-type receptors in other taxa, which is reflected in their position peripheral to the main cluster of sNPF/PrRP-type receptors in the CLANS. Accordingly, it is noteworthy that in the phylogenetic tree (*Figure 4*) there is a long branch leading to the echinoderm receptor clade and likewise nematode sNPF-type receptors also have long branches (*Figure 4*). Nevertheless, collectively our sequence analysis indicates that the echinoderm receptors are orthologs of sNPF/PrRP-type receptors. Therefore, it was of interest to determine if echinoderm PrRP-type neuropeptides act as ligands for sNPF/PrRP-type receptors in this phylum.

Here we show that the *A. rubens* PrRP-type neuropeptide ArPrRP (pQDRSKAMQAERTG QLRRLNPRF-NH$_2$) is a potent ligand for the *A. rubens* sNPF/PrRP–type receptor Ar-sNPF/PrRPR (*Figure 5*). These findings demonstrate for the first time the existence and molecular identity of a PrRP-type signalling system in an echinoderm. Furthermore, our identification of orthologs of ArPrRP and Ar-sNPF/PrRPR in other echinoderms, including for example the sea urchin *S. purpuratus*, demonstrates the conservation of this signalling system in this phylum. In addition, our comparative analysis of sequence data has also enabled identification of genes/transcripts encoding PrRP-type neuropeptides in the hemichordate *S. kowalevskii* and the cephalochordate *B. floridae* (*Figure 1*).

## Reconstructing the evolutionary history of PrRP/sNPF-type neuropeptide signalling

Previous studies have concluded that sNPF-type signalling is paralogous to NPY/NPF-type signalling in protostomes (**Nässel and Wegener, 2011**) and that PrRP-type signalling is paralogous to NPY/NPF-type signalling in vertebrates (**Lagerström et al., 2005**). Evidence that the PrRP-type and sNPF-type signalling systems may be orthologous has also been reported previously (**Jékely, 2013**), but this hypothesis has not been tested experimentally. Our discovery of a starfish PrRP-type neuropeptide that acts as a ligand for a starfish ortholog of sNPF-type receptors is important because it provides a missing link for reconstruction of the evolutionary history of PrRP/sNPF-type neuropeptide signalling (*Figure 6*).

Comparison of the sequences of vertebrate PrRP-type neuropeptides and protostome sNPF-type neuropeptides reveals low levels of sequence similarity, which no doubt in part explains why PrRP-type and sNPF-type neuropeptides have not been recognised as orthologs. In *Figure 4—figure*

*supplement 1* we illustrate this in an alignment of the echinoderm PrRP-type neuropeptides and protostome sNPF-type neuropeptides, with sequence identity restricted to a few residues in the C-terminal regions of these peptides. This contrasts with the higher levels of sequence similarity shared between echinoderm PrRP-type neuropeptides and vertebrate PrRP-type neuropeptides, as shown in *Figure 1A*. Furthermore, echinoderm PrRP-type precursors are similar to chordate PrRP-type precursors in containing a single long neuropeptide, whereas protostome sNPF-type precursors typically contain multiple smaller neuropeptides. Thus, there is little evidence of orthology from comparison of echinoderm PrRP-type and protostome sNPF-type neuropeptide, precursor and gene sequences. Consequently, our conclusion that the echinoderm PrRP-type peptides are orthologs of protostome sNPF-type peptides is principally based on the orthology of their receptors (*Figure 3*) and our experimental demonstration that a PrRP-like peptide (ArPrRP) acts as a ligand for a sNPF/PrRP-type receptor (Ar-sNPF/PrRPR) in the starfish *A. rubens* (*Figure 5*). It is important to note, however, that this is not unprecedented in investigations of the evolution of neuropeptide signalling. Thus, whilst the sequences of some neuropeptides and neuropeptide precursors are highly conserved throughout the Bilateria, others are so divergent that they can be unrecognisable as orthologs. An example of the former are vasopressin/oxytocin (VP/OT)-type neuropeptides and precursors. An example of the latter are neuropeptide-S (NPS)/crustacean cardioactive peptide (CCAP)-type neuropeptides and precursors, which are paralogs of VP/OT-type neuropeptides and precursors (*Semmens et al., 2015*). Thus, by way of comparison, NPY/NPF-type neuropeptides are similar to VP/OT-type neuropeptides in exhibiting a high level of sequence conservation throughout the Bilateria. Conversely, PrRP/sNPF-type neuropeptides are similar to NPS/CCAP-type neuropeptides in being highly divergent, with neuropeptides in protostomes and deuterostomes exhibiting modest sequence similarity.

The discovery of PrRP/sNPF-type signalling in echinoderms has provided a unique opportunity to speculate on the ancestral characteristics of this signalling system in Urbilateria. It is noteworthy that, by comparison with the protostome sNPF-type peptides, the echinoderm PrRP-type peptides have more features in common with the paralogous NPY/NPF-type peptides. PrRP-type peptides are not as long as NPY/NPF-type peptides but they are nevertheless much longer than protostome sNPF-type peptides. Furthermore, it was the sequence similarity that echinoderm PrRP-type peptides share with NPY/NPF-type peptides that originally facilitated their discovery (*Zandawala et al., 2017*). Additionally, the structure of the PrRP-type precursors is similar to NPY/NPF-type precursors because the neuropeptide is located immediately after the signal peptide, whereas this is not a feature of protostome sNPF-type precursors. Based on these observations, we propose that PrRP-type peptides and precursors may more closely resemble the ancestral characteristics of the PrRP/sNPF type signalling system in Urbilateria. Furthermore, we speculate that the common ancestor of the paralogous NPY/NPF-type and PrRP/sNPF-type neuropeptide precursors may have been similar to NPY/NPF-type precursors with respect peptide, precursor and gene structure. Then, following gene duplication, these ancestral characteristics were retained in the paralog that gave rise to the bilaterian NPY/NPF-type peptides/precursors. In contrast, the paralog that gave rise to PrRP/sNPF-type signalling diverged from the ancestral condition. However, the extent of divergence varies in the deuterostome and protostome lineages. In deuterostomes, the PrRP-type peptides/precursors have many NPY/NPF-type characteristics and we conclude that this reflects less divergence from the proposed ancestral condition. Conversely, in the protostomes, the sNPF-type peptides/precursors exhibit little similarity with NPY/NPF-type peptides/precursors and we conclude that this reflects more divergence from the proposed ancestral condition.

In conclusion, our discovery of a PrRP/sNPF-type signalling system in echinoderms has provided a missing link that unites PrRP-type peptides in vertebrates and sNPF-type peptides in protostomes as members of a bilaterian family of neuropeptides, as illustrated in *Figure 6*. This represents an important advance in our knowledge of neuropeptide signalling systems in the Bilateria and illustrates the value of insights from echinoderms in enabling reconstruction of the evolutionary history of neuropeptides.

## Materials and methods

### Key resources table

| Reagent type (species) or resource | Designation | Source or reference | Identifiers | Additional information |
|---|---|---|---|---|
| Recombinant DNA reagent | pBluescript II KS (+) plasmid (cloning vector) | Invitrogen | Cat# K280002 | |
| Recombinant DNA reagent | pcDNA3.1(+) with neomycin selectable marker (mammalian expression vector) | Invitrogen | Cat# V790-20 | |
| Commercial assay, kit | Lipofectamine 3000 | Invitrogen | Cat# L3000015 | |
| Transfected construct (*Asterias rubens*) | *Asterias rubens* sNPF/PrPR receptor cDNA cloned into an expression vector | This paper | Genbank: MH807444 | Cloned in the plasmid pcDNA3.1+ from Invitrogen |
| Transfected construct (*Aequorea victoria*) | Chimeric green fluorescent protein-aequorin fusion protein (G5A) | *Baubet et al., 2000* | N/A | Cloned into the pEGFP-C1 vector (CLONTECH) |
| Transfected construct (*Homo sapiens*) | Human guanine nucleotide binding protein, alpha 15 (16) (Gq class) | cDNA resource center | Cat# GNA1500000 | HGNC ID:4383 Human GNA15 cloned into the plasmid pcDNA3.1+. |
| Cell line (*Cricetulus griseus*) | Chinese hamster ovary cells (CHO-K1) | Sigma-Aldrich | RRID:CVCL_0214 | Cat. No. 85051005 |
| Software, algorithm | Prism | GraphPad | Version 7.0 | |
| Software, algorithm | Sequest Proteome Discoverer | Thermo Fisher Scientific | Version 2.2 | |
| Software, algorithm | Scaffold | Proteome Software | Version 4.8.4 | |

## Animals

Starfish (*Asterias rubens*) were obtained from a fisherman based at Whitstable (Kent, UK). They were then maintained in a circulating seawater aquarium at ~11°C in the School of Biological and Chemical Sciences at Queen Mary University of London and were fed on mussels (*Mytilus edulis*) collected near Margate (Kent, UK).

## Cloning and sequencing of a cDNA encoding the precursor of an *A. rubens* NPY/NPF/PrRP-like peptide

A transcript encoding the *A. rubens* precursor of an NPY/NPF-like peptide was reported previously (GenBank: MK033631) (*Zandawala et al., 2017*). However, in this paper we show that the NPY/NPF-like peptide derived from this precursor shares more sequence similarity with PrRP-type peptides. A cDNA containing the complete open reading frame of the precursor was amplified by PCR using *A. rubens* radial nerve cord cDNA, the forward primer AAGTCAAAAGGCGAGCAAGA, the reverse primer AAAGGGATGTGGTGTTGGTG and Q5 polymerase (NEB; Cat. No. M0491S). The PCR products were ligated into the pBluescript II KS (+) vector (Invitrogen; Cat. No. K280002) that had been cut previously with the restriction enzyme *EcoRV* by performing blunt-end ligation with T4 DNA ligase (NEB; Cat. No. M0202S). The cloning was confirmed by restriction enzyme digestion and sequencing (TubeSeq service; Eurofins Genomics).

## Structural characterisation of the *A. rubens* NPY/NPF/PrRP-like peptide using mass spectrometry

After confirming the nucleotide sequence of the *A. rubens* precursor of a NPY/NPF/PrRP-like peptide by cloning and sequencing, mass spectrometry was used to determine the mature structure of the peptide. The methods employed, including extraction of peptides from *A. rubens* radial nerve cords, treatment of samples, equilibration of columns, reverse phase chromatography for the initial separation and injection into a Orbitrap-Fusion (ThermoScientific) for tandem mass spectrometry

(MS/MS), were performed using a previously reported protocol for the identification of the starfish neuropeptides (*Lin et al., 2017*). The methods employed for data analysis are described below. Mass spectra were searched using Sequest Proteome Discoverer (Thermo Fisher Scientific, v. 2.2) against a database comprising forty-three different precursor proteins identified by analysis of *A. rubens* neural transcriptome data, including the *A. rubens* ArPRP precursor and all proteins in Gen-Bank from species belonging to the Asteriidae family and the common Repository of Adventitious Proteins Database (http://www.thegpm.org/cRAP/index.html). Theoretical peptides were generated allowing up to two missed cleavages and variable modifications, including amidation (−0.98402) of C-terminal glycines and pyroglutamate (−17.02655) of N-terminal glutamines, and oxidation of methionine (+15.99). Precursor mass tolerance was 10 ppm and fragment ions were searched at 0.8 Da tolerances. Results from Discoverer were collated and annotated in Scaffold version 4.8.4 (Proteome Software).

## Sequence alignment of echinoderm NPY/NPF/PrRP-like peptides with NPY/NPF-type peptides, PrRP-type peptides, and sNPF-type peptides from other taxa

The amino acid sequences of echinoderm NPY/NPF/PrRP-like peptides were aligned with the sequences of NPY/NPF-type peptides, PrRP-type peptides and sNPF-type peptides from a variety of bilaterian species (see *Figure 1—source data 1* and *Figure 4—figure supplement 1—source data 1* for lists of the sequences). To identify candidate ligands for PrRP-type receptors in the cephalochordate *B. floridae* and the hemichordate *S. kowalevskii*, we analysed transcriptomic and genomic sequence data for these species (*Putnam et al., 2008*; *Simakov et al., 2015*). The data analysed also included a list of predicted *S. kowalevskii* proteins kindly provided to O. Mirabeau by Dr. R.M. Freeman (Harvard Medical School, USA). The methods employed to identify candidate neuropeptide precursors have been reported previously (*Mirabeau and Joly, 2013*) but here we had the more specific objective of identifying proteins with an N-terminal signal peptide followed by a neuropeptide with a predicted C-terminal RFamide or RYamide motif. This resulted in discovery of one candidate PrRP-type precursor in the cephalochordate *B. floridae* and two candidate PrRP-type precursors in the hemichordate *S. kowalevskii*.

Alignments were performed using MAFFT version 7 (5 iterations, substitution matrix; BLOSUM62) and then manually curated. Highlighting of the conserved residues was done using BOXSHADE (www.ch.embnet.org/software/BOX_form.html) with 50% conservation as the minimum for highlighting. Finally, the sequences were highlighted in phylum-specific or superphylum-specific colours: dark blue (Echinodermata), light blue (Hemichordata), purple (Chordata), orange (Platyhelminthes), red (Lophotrochozoa), yellow (Priapulida), green (Arthropoda), grey (Nematoda).

## Comparison of the exon/intron structure of genes encoding NPY/NPF/PrRP-like peptides in echinoderms and genes encoding NPY/NPF-type peptides, PrRP-type peptides and sNPF-type peptides in other taxa

The sequences of transcripts and genes encoding precursors of echinoderm precursors of NPY/NPF/PrRP-like peptides and precursors of NPY/NPF-type, PrRP-type and sNPF-type peptides from other taxa were obtained from GenBank. The sequence of a predicted transcript encoding a second *S. kowalevskii* precursor (Skow2) of a PrRP-like peptide was determined based on a GenScan prediction (*Burge and Karlin, 1997*; *Burge and Karlin, 1998*) from scaffold 51909 (GenBank accession number NW_003156735.1). See *Figure 2—source data 1* and *Figure 4—figure supplement 2—source data 1* for a list of the transcript and gene sequences analysed. The online tool Splign (*Kapustin et al., 2008*) (https://www.ncbi.nlm.nih.gov/sutils/splign/splign.cgi) was employed to determine the exon/intron structure of genes and schematic figures showing gene structure were generated using IBS 1.0 (*Liu et al., 2015*).

## Identification of a candidate receptor for the NPY/NPF/PrRP-like peptide in *A. rubens* and analysis of its relationship with NPY/NPF/PrRP/sNPF-type receptors in other taxa

To identify a candidate receptor for the *A. rubens* NPY/NPF/PrRP-like peptide, *A. rubens* neural transcriptome sequence data were analysed using the BLAST server SequenceServer (*Priyam et al.,*

*2015*), submitting NPY-type receptors from *H. sapiens* (GenBank NP_000900.1, NP_000901.1, NP_001265724.1), an NPF-type receptor from *D. melanogaster* (GenBank AAF51909.3), a PrRP-type receptor from *H. sapiens* (NP_004239.1) and sNPF-type receptors from *D. melanogaster* (GenBank; NP_524176.1) and *C.gigas* (GenBank XP_011451552.1) as query sequences. A transcript (contig 1120879) encoding a 386-residue protein (http://web.expasy.org/translate/) was identified as the top hit in all BLAST searches and this has been deposited in GenBank under the accession number MH807444. The protein sequence was also analysed using Protter V1.0 (*Omasits et al., 2014*). Using BLAST, homologs of the *A. rubens* protein were identified in other echinoderms for which genome sequences are available, including the starfish *Acanthaster planci* (XP_022101544.1), the sea urchin *Strongylocentrotus purpuratus* (XP_003725178.1) and the sea cucumber *Apostichopus japonicus* (PIK36230.1). Furthermore, no other NPY/NPF/PrRP/sNPF-type receptors were identified in these species.

To investigate the relationship of the echinoderm receptors with neuropeptide receptors from other bilaterians, a database of receptor sequences was generated that included NPY/NPF-type, PrRP-type, sNPF-type, tachykinin-type, luqin-type and GPCR83-type receptors (the latter three receptor types being included as outgroups), including representative species from the phyla Chordata, Hemichordata, Echinodermata, Mollusca, Annelida, Platyhelminthes, Nematoda, Priapulida, and Arthropoda (see *Figure 3* – source data for a list of the sequences used). A cluster-based analysis of the receptor sequences was performed using CLANS (*Frickey and Lupas, 2004*). An all-against-all BLAST was performed using the scoring matrix BLOSUM62 and linkage clustering was performed with an e-value of 1e-68 to identify coherent clusters. The clustering was first performed in 3D and then the map was collapsed to 2D to enable generation of the diagram shown in *Figure 3* (see *Figure 3* – source data for a list of sequences used). Using the same receptor sequences, a phylogenetic tree was generated using the maximum-likelihood method. Receptor sequences were aligned using MUSCLE in the online tool NGPhylogeny (iterative, 16 iterations, UPGMB as clustering method) (*Edgar, 2004*; *Lemoine et al., 2019*) and the alignment was automatically trimmed using trimAL with automatic selection of trimming method using the online tool NGPhylogeny (*Capella-Gutierrez et al., 2009*). The trimming contained a total of 239 residues that were used to generate the maximum-likelihood tree using W-IQ-tree online version 1.0 (the model was automatically selected, being LG+G+I+F the chosen substitution model, branch tests used were ultrafastbootstrap 1000 replicates and SH-aLRT 1000 replicates) (*Trifinopoulos et al., 2016*). The sequence database used for this tree, together with the trimmed alignment, and the raw tree are available at Zenodo (https://zenodo.org/record/3837351).

## Cloning a candidate receptor for the NPY/NPF/PrRP-like peptide in *A. rubens*

To enable the pharmacological characterisation of a candidate receptor for the *A. rubens* NPY/NPF/PrRP-like peptide, a cDNA encoding this receptor was cloned into the eukaryotic expression vector pcDNA 3.1(+) (Invitrogen; Cat. No. V790-20). To facilitate expression of the cloned receptor, the forward primer included a partial Kozak consensus sequence (ACC) and a sequence corresponding to the first 15 bases of the open reading frame of contig 1120879 (ACCATGCAGATGACAACC) and the reverse primer consisted of a stop codon and a sequence reverse complementary to the 3' region of the open reading frame of contig 1120879 (GCGTCACATAGTGGTATCATG). PCR was performed using the forward primer and reverse primers, *A. rubens* radial nerve cord cDNA and Q5 polymerase (NEB; Cat. No. M0491S). PCR products were ligated into the pcDNA 3.1(+) vector that had been cut previously with the restriction enzyme *EcoRV* by performing blunt-end ligation with T4 DNA ligase (NEB; Cat. No. M0202S). Successful ligation and the direction of the insert was determined by restriction enzyme digestion and sequencing (TubeSeq service; Eurofins Genomics).

## Cell lines and pharmacological characterisation of a candidate receptor for the NPY/NPF/PrRP-like peptide in *A. rubens*

Chinese hamster ovary (CHO)-K1 cells stably expressing the calcium sensitive apoaequorin-GFP fusion protein (G5A) (*Baubet et al., 2000*) were used here for receptor assays. These cells have been used previously for neuropeptide receptor deorphanisation (*Bauknecht and Jékely, 2015*) and were generously supplied to us by Dr Gáspár Jékely (University of Exeter). The cell line was

generated using the CHO-K1 cell line from Sigma-Aldrich (85051005), which is certified by the European Collection of Authenticated Cell Cultures (ECACC). Following transfection with a plasmid encoding G5A, cells were selected for stable transfection using Geneticin G418 sulfate (Thermo Fisher Scientific, Cat. No. 10131035). The methods we used for cell culture and receptor assays have been described previously (*Yañez-Guerra et al., 2018*). Upon reaching a confluency of approximately 80%, cells were transfected with a plasmid containing the Ar-sNPF/PrRP receptor cDNA and a plasmid containing the promiscuous $G\alpha-16$ protein that can couple a wide range of GPCRs to the phospholipase C signalling pathway. The transfection was achieved using 5 µg of each plasmid and 10 µl of the transfection reagents P3000 and Lipofectamine 3000 (Thermo Fisher Scientific; Cat. No. L3000008), as recommended by the manufacturer. It was not possible to authenticate the CHO-K1 (G5A) cells or test the cells for mycoplasma contamination at the time of manuscript submission due to laboratory closure during the COVID-19 pandemic.

After transfection with the *A. rubens* receptor, cells were exposed to the *A. rubens* NPY/NPF/PrRP-like peptide pQDRSKAMQAERTGQLRRLNPRF-$NH_2$ (custom synthesised by Peptide Protein Research Ltd., Fareham, UK), which was diluted in DMEM/F12 Nutrient Mixture medium at concentrations ranging from $10^{-14}$ M to $10^{-5}$ M in clear bottom 96-well plates (Sigma-Aldrich; Cat. No. CLS3603-48EA). Luminescence was measured over a 30 s period using a FLUOstar Omega Plate Reader (BMG LABTECH; FLUOstar Omega Series multi-mode microplate reader) and data were integrated over the 30 s measurement period. For each concentration, measurements were performed in triplicate, and the average of each was used to normalise the responses. The responses were normalised to the maximum luminescence measured in each experiment (100% activation) and to the background luminescence with the vehicle media (0% activation). Dose-response curves were fitted with a four-parameter curve and $EC_{50}$ values were calculated from dose–response curves based on at least three independent transfections using Prism 6 (GraphPad, La Jolla, USA).

## Acknowledgements

The work reported in this paper was supported by grants from the BBSRC awarded to MRE (BB/M001644/1) and AMJ (BB/M001032/1). LAYG was supported by a PhD studentship awarded by the Mexican Council of Science and Technology (CONACyT studentship no. 418612) and Queen Mary University of London and by a Leverhulme Trust grant (RPG-2016–353) awarded to MRE. XZ was supported by a PhD studentship awarded by the China Scholarship Council and Queen Mary University of London. We are grateful to Gáspár Jékely (University of Exeter) for providing the CHO-K1 (G5A) cell line used here for receptor deorphanisation assays.

## Additional information

### Funding

| Funder | Grant reference number | Author |
| --- | --- | --- |
| Biotechnology and Biological Sciences Research Council | BB/M001644/1 | Maurice R Elphick |
| Biotechnology and Biological Sciences Research Council | BB/M001032/1 | Alexandra M Jones |
| CONACyT | CONACyT studentship no. 418612 | Luis Alfonso Yañez-Guerra |
| Leverhulme Trust | RPG-2016-353 | Maurice R Elphick |
| China Scholarship Council | | Xingxing Zhong |

The funders had no role in study design, data collection and interpretation, or the decision to submit the work for publication.

### Author contributions

Luis Alfonso Yañez-Guerra, Xingxing Zhong, Conceptualization, Data curation, Formal analysis, Validation, Investigation, Visualization, Methodology, Writing - original draft, Writing - review and

editing; Ismail Moghul, Software, Formal analysis, Writing - review and editing; Thomas Butts, Super-vision, Writing - review and editing; Cleidiane G Zampronio, Formal analysis, Investigation, Visualiza-tion, Methodology, Writing - review and editing; Alexandra M Jones, Formal analysis, Supervision, Funding acquisition, Investigation, Visualization, Writing - review and editing; Olivier Mirabeau, Soft-ware, Formal analysis, Investigation, Methodology, Writing - review and editing; Maurice R Elphick, Conceptualization, Resources, Formal analysis, Supervision, Funding acquisition, Writing - original draft, Project administration, Writing - review and editing

## Author ORCIDs

Luis Alfonso Yañez-Guerra (iD) https://orcid.org/0000-0002-2523-1310
Ismail Moghul (iD) https://orcid.org/0000-0003-3653-2327
Maurice R Elphick (iD) https://orcid.org/0000-0002-9169-0048

## Decision letter and Author response

Decision letter https://doi.org/10.7554/eLife.57640.sa1
Author response https://doi.org/10.7554/eLife.57640.sa2

## Additional files

### Supplementary files

• Transparent reporting form

### Data availability

All data generated or analysed during this study are included in the manuscript and supporting files.

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
