## [Decision Letter]

**Acceptance summary:**

In this manuscript, the authors explore the transcriptome of *A. rubens* and the genomes of three other echinoderms to identify a neuropeptide signaling system related to NPF/NPY/sNPF/PrRP in Echinoderms. They use various approaches, including sequence alignments, intron-exon structures, cluster analysis, phylogenetic approaches and pharmacology to show that echinoderms seem not to have NPY/NPF signaling systems, and instead have a single sNPF/PrRP type receptor and a corresponding PrRP like ligand. Their results suggest that there has been a loss of the NPY/NPF system in echinoderms.

**Decision letter after peer review:**

[Editors’ note: the authors submitted for reconsideration following the decision after peer review. What follows is the decision letter after the first round of review.]

Thank you for submitting your work entitled "Urbilaterian origin and evolution of sNPF-type neuropeptide signalling" for consideration by *eLife*.

In this manuscript, the authors investigate a family of peptide signalling systems – the Neuropeptide Y/F (NPF/Y), and the related short Neuropeptide (sNPF) – in the Echinoderm, *Asteria rubens*, which is a deuterostome.

Although the genome of *A. rubens* has not been sequenced, a transcriptome of the radial nerve cord is available. Using reference sequences, the authors query this transcriptome for NPY-like ligands and receptors to identify a single putative candidate for each.

For the receptor, they perform phylogenetic analysis (with receptor sequences from Echinoderm species that have been sequenced). Surprisingly, the Echinoderm candidate receptor did not cluster with the NPY/F. Instead it clustered with the sNPF type receptor family. This is interesting because it's been proposed that sNPF type signalling systems are restricted to protostomes and is absent from deuterostomes.

Presumably due to the complexity of the peptides and precursors of peptides, such phylogenetic analysis is not possible for the ligands. So, the authors rely on peptide sequence alignments and structure of introns and exons of the precursor genes to draw inferences about evolutionary conservancy. Based on this, they find that their putative ligand is most similar to a third family of peptide signalling systems – the chordate PrRPs. This is also interesting, because PrRPs have, until now, been thought to be a chordate-specific signalling system. Interestingly, their putative receptor, had not clustered with the PrRP type receptors.

The authors also show that the PrRP-like ligand is capable of binding to and activating the sNPF-like receptor in a heterologous cell system.

Based on this these findings (and other considerations), the authors propose that the Echinoderm sNPF system is an ancestral signalling system that was lost in the chordates.

There are many substantial concerns given below. As the authors can see, addressing these will make this a very different work.

Reviewer #1:

Since the manuscript is entitled “Urbilaterian origin and evolution of sNPF-type neuropeptide signaling”, I expected that the authors would have explored the available echinoderm genomes for NPY/F, sNPF and PrRP type receptors in an unsupervised manner. As I understand it, the authors have picked a single receptor candidate based on a query of the transcriptome against Human and Fly NPY and sNPF receptors. Does this targeted approach miss important relationships of signaling systems in Echinoderms?

Despite this, I do agree that their data indicates that the sNPF signaling system might have existed prior to the protostome/deuterostome split. I also find it very interesting that the authors show a seemingly hybrid peptide signaling system – a PrRP-type ligand binding an sNPF-type receptor. This could suggest that the two systems were present before the deuterostome split, and while the PrRP system flourished in most deuterostome lineages, sNPF was lost. So, it would really be very interested to know what other receptors (and ligands) of the NPY/F, sNPF and PrRP families are present in the genomes of other Ambulacrarians.

If the authors feel that such analysis is beyond the scope of this manuscript, maybe they could consider changing to title to reflect that they have described an sNPF-PrRP hybrid peptidergic signaling system in *A. rubens*. And in this scenario they could maybe describe the evolutionary implications of their findings in the Discussion section.

Finally, may I please request the authors revisit their writing? While their findings are very interesting, the manuscript is difficult to read: take, for example, the sentence “Furthermore, alignment of NPY-type peptides and precursors from vertebrates with

NPF-type and sNPF-type peptides and precursors from protostomes revealed that whilst NPF peptides are clearly closely related (orthologous) to vertebrate NPY peptides, sNPF peptides and precursors exhibit too many differences to be considered orthologs of NPY/NPF-type peptides and precursors (Nässel and Wegener, 2011).”. The story is also presented as a back-and-forth between receptor and ligand, which adds to the difficulty. And finally, the Discussion contains large pieces of writing that belong to the Results section – some of which is discussed in the Results section, but some that are not.

Reviewer #2:

I would be more inclined to this work if it were in the short communications category. Also, I am not qualified to assess the details of their methods, but I was not convinced by their arguments that echinoderms have a sNPF system. In my view, the ligands and the receptors lead to opposite conclusions. They have not convinced me why they should favor one over the other. My comments are:

The manuscript by Dr. Yanez Guera et al., describes the discovery of a novel sNPF-like signaling system in echinoderms and uses this information to devise a scheme for the evolution of the NPY/NPF versus sNPF signaling in the bilateria. I found the paper to be difficult to read because it was a hybrid between a research report and a review. Also, it took a historical approach to presenting the data, which made it difficult to follow and to understand why certain topics were being covered.

One example is their focus in the beginning on the NPY/NPF system. This was obviously relevant when the authors first thought that their peptide was an echinoderm member of this family and they referred to it as ArNPYLP. However, neither the gene structure nor the peptide sequence supported placement in this family. A strong point to the paper is that to better understand their starfish peptide they sought its receptor. They could not find NPY/NPF-like receptors in starfish but they did find a receptor related to protostome sNPF receptor that did bind the peptide with sub-nanomolar affinity. At this point they then changed the name of the peptide to Ar-sNPF. I found this name switch in mid-paper to be quite confusing!

The main point of the paper is that the sNPF system is not confined to the protostomes as previously thought. However, I think that the data supporting that echinoderms have a sNPF signaling system is weak. They show that the gene structure of their Ar-sNPF has no resemblance to known sNPF genes from protostomes (Figure ). In comparing Ar-sNPF to other sNPF's the authors speak of "modest" similarity, but most of the similarity is in the C-terminal RFamide which is shared by many other peptide families. Indeed, as they then show, the better sequence match is with PrRP-like peptides of hemicordates and the PrRP-type peptides of chordates. They would be on firmer ground calling the starfish molecule a PrRP-like peptide. The reason for concluding that the starfish peptide is used in a sNPF-like system then rested with the receptor. Jekely (2013) concludes that the receptors for the PrRP and sNPF peptides cluster together. This differs from Figure 3 of this paper, and it is not clear to me which interpretation is better supported. In their Figure, the echinoderm receptors are an out group to the rest of the sNPF-type receptors. I do not know the level of confidence in placing them with the sNPF type receptors rather than being over with the hemicordates and the PrRP-type receptors. As it stands right now, the hemicordates and the echinoderms have similar peptides but they work through two different types of receptors.

[Editors’ note: further revisions were suggested prior to acceptance, as described below.]

Thank you for resubmitting your work entitled "Echinoderms provide missing link in the evolution of PrRP/sNPF-type neuropeptide signalling" for further consideration by *eLife*. Your revised article has been evaluated by K VijayRaghavan as the Senior Editor and Reviewing Editor, and two reviewers.

The manuscript has been improved but there are some remaining issues that need to be addressed before acceptance, as outlined below:

Summary:

In its revised form, the work warrants publication in *eLife* after revision. The work uses both genomic and experimental approaches to resolve the evolutionary history and relationships of an ancient and diverse group of neuropeptides – the NPF/NPY/sNPF/PrRP peptides. Sequence alignments, intron-exon structures, cluster analysis, phylogenetic approaches and pharmacology are used to show that echinoderms seem not to have NPY/NPF signalling systems, and instead have a single sNPF/PrRP type receptor and a corresponding PrRP like ligand.

The relationship of the NPF and NPY neuropeptides is well established, but a presumed neuropeptide of this group from Echinoderms seemed problematic. The authors provide evidence that this peptide is actually more related to the chordate PrRP neuropeptides but its possible receptor seems to come from the sNPF family. They then show that the starfish peptide (ArPrRP) is a potent ligand for this putative sNPF receptor. This provides a way of reconstructing the evolutionary history of PrRP/sNPF signaling. Their results suggest that there has been a loss of the NPY/NPF system in echinoderms.

We appreciate the effort the authors have taken to address many of the concerns we raised earlier. The manuscript is well written, and the data clearly presented.

Substantive Concerns:

The inconsistency in receptor identification between the two methods and the long branches in the phylogenetic analysis are of concern. We wonder if the authors have looked at the intron-exon structure of the receptors?

The paper does not require and more experiments or analysis but there are a couple of points that the authors might address.

1) In the Discussion, the section "Discovery of a PrRP/sNPF-type neuropeptide signaling system in echinoderms" is long and simply repeats the data that are presented in the Results. It is not really Discussion material and it could be dropped or greatly condensed without affecting the paper.

2) In comparing the echinoderm peptides to PrRP-type or NPY/NPF type peptides, the authors point out that there are "thirteen other residues in the echinoderm peptides that are identical to equivalently positioned residues in at least one of the chordate or hemicordate PrRPs, as highlighted by the asterisks" In part B of Figure 1, using the same criteria for comparing the starfish peptide to known NPY/NPF type peptides, almost every residue is denoted by an asterisk. It is important to point out highly conserved residues, but the significance of the co-occurrence of a particular residue in a single other species is questionable. We suggest dropping that part of the analysis.

---

## [Author Response]

[Editors’ note: the authors resubmitted a revised version of the paper for consideration. What follows is the authors’ response to the first round of review.]

The feedback from the reviewers was very helpful to us and accordingly we have produced a substantially revised manuscript that addresses all of the concerns raised by the reviewers. In particular we have:

a) obtained new data using CLANS that revises and strengthens the conclusions of the paper;

b) changed the structure and logic of the paper;

c) removed data that was non-essential for the main objective of the study;

d) changed the title to reflect the above changes.

Reviewer #1:Since the manuscript is entitled “Urbilaterian origin and evolution of sNPF-type neuropeptide signaling”, I expected that the authors would have explored the available echinoderm genomes for NPY/F, sNPF and PrRP type receptors in an unsupervised manner. As I understand it, the authors have picked a single receptor candidate based on a query of the transcriptome against Human and Fly NPY and sNPF receptors. Does this targeted approach miss important relationships of signaling systems in Echinoderms?

It is unfortunate that the reviewer gained the impression that we did not explore echinoderm genomes for NPY/F, sNPF and PrRP type receptors in an unsupervised manner. In fact our analysis of sequence data included echinoderm species for which genome sequences are available, the starfish Acanthaster planci, the sea urchin Strongylocentrotus purpuratus and the sea cuumber Apostichopus japoncus , and we found that in these species only a single NPY/NPF/PrRP/sNPF-type receptor is present, consistent with our analysis of *A. rubens* transcriptome sequence data. We have modified the manuscript accordingly to provide greater clarity on this point:

In the Results section of the manuscript we now state:

“Furthermore, homologs of the *A. rubens* protein encoded by contig 1120879 were also identified in other echinoderms for which genome sequences have been obtained, including the starfish A. planci, the sea urchin S. purpuratus and the sea cucumber A. japonicus , and importantly no other NPY/NPF/PrRP/sNPF-type receptors were identified in these species.”

In the Materials and methods section of the manuscript we now state:

“Using BLAST, homologs of the *A. rubens* protein were identified in other echinoderms for which genome sequences are available, including the starfish Acanthaster planci (XP_022101544.1), the sea urchin Strongylocentrotus purpuratus (XP_003725178.1) and the sea cucumber Apostichopus japonicus (PIK36230.1). Furthermore, no other NPY/NPF/PrRP/sNPF-type receptors were identified in these species.”

Despite this, I do agree that their data indicates that the sNPF signaling system might have existed prior to the protostome/deuterostome split. I also find it very interesting that the authors show a seemingly hybrid peptide signaling system – a PrRP-type ligand binding an sNPF-type receptor. This could suggest that the two systems were present before the deuterostome split, and while the PrRP system flourished in most deuterostome lineages, sNPF was lost. So, it would really be very interested to know what other receptors (and ligands) of the NPY/F, sNPF and PrRP families are present in the genomes of other Ambulacrarians.

In the revised version of the paper, we conclude that PrRP-type signalling in vertebrates and sNPF-type signalling in protostomes are orthologous and our discovery of a PrRP-like peptide that acts as a ligand for a sNPF/PrRP-type receptor in an echinoderm is a key “missing link” that supports this conclusion. Based on our analysis of genome sequence data we conclude that in the ambulacrarian branch of the deuterostomes, sNPF/PrRP-type signalling has been retained in echinoderms and hemichordates. In contrast, the paralogous NPY/NPF-type signalling system has been lost in echinoderms but retained in hemichordates. This is summarised in Figure 6.

If the authors feel that such analysis is beyond the scope of this manuscript, maybe they could consider changing to title to reflect that they have described an sNPF-PrRP hybrid peptidergic signaling system in A. rubens. And in this scenario they could maybe describe the evolutionary implications of their findings in the Discussion section.

We have generated new data using CLANS to analyse relationships of NPY/NPF/sNPF/PrRP-type receptors in the Bilateria (see new Figure 3). A previous study using this methodology (Jekely, 2013) provided important evidence that PrRP-type receptors and sNPFtype receptors are orthologous. A novelty of our analysis is the inclusion of several echinoderm receptor sequences and, interestingly, the echinoderm receptors showed stronger connections with the sNPF/PrRP receptor cluster (Figure 3) than with the NPY/NPF receptor cluster. It is noteworthy, however, that whilst strong connections between the echinoderm receptors and PrRP/sNPF-type receptors in other taxa can be seen using CLANS, the lines linking to the echinoderm receptors are quite long (Figure 3). This suggests that the echinoderm receptors are orthologs of PrRP/sNPF-type receptors but have undergone sequence divergence.

The reviewer’s excellent suggestion that we change the title of the paper to reflect our discovery of a “sNPF-PrRP hybrid peptidergic signaling system” in echinoderms is entirely consistent with the findings of the revised paper and accordingly we have changed the title of the paper to:

“Echinoderms provide missing link in the evolution of PrRP/sNPFtype neuropeptide signalling”

Accordingly, we have also produced a revised summary figure (new Figure 6) that illustrates the phylogenetic distribution and evolution of the sNPF/PrRP-type and NPY/NPF-type signalling systems.

Finally, may I please request the authors revisit their writing? While their findings are very interesting, the manuscript is difficult to read: take, for example, the sentence “Furthermore, alignment of NPY-type peptides and precursors from vertebrates withNPF-type and sNPF-type peptides and precursors from protostomes revealed that whilst NPF peptides are clearly closely related (orthologous) to vertebrate NPY peptides, sNPF peptides and precursors exhibit too many differences to be considered orthologs of NPY/NPF-type peptides and precursors (Nässel and Wegener, 2011).”. The story is also presented as a back-and-forth between receptor and ligand, which adds to the difficulty. And finally, the Discussion contains large pieces of writing that belong to the Results section – some of which is discussed in the Results section, but some that are not.

We agree that the original version of our paper was too long and complex in structure. Accordingly, we have completely changed the structure of the paper so that the paper begins by highlighting the sequence similarity that the echinoderm neuropeptides share with vertebrate PrRPs, rather than placing this at the end. This then changes the logic and structure of the paper in a way that makes it much easier to follow. Furthermore, we have completely removed from the paper our characterisation of a molluscan sNPF-type receptor, data that were referred to in the Discussion section of the original paper but which is not essential for the main objectives the study.

Reviewer #2:I would be more inclined to this work if it were in the short communications category.

We agree that the original version of the paper was too long (10,230 words including Materials and methods) and we have now produced a revised version that is much shorter (7,103 words, including Materials and methods).

Also, I am not qualified to assess the details of their methods, but I was not convinced by their arguments that echinoderms have a sNPF system. In my view, the ligands and the receptors lead to opposite conclusions. They have not convinced me why they should favor one over the other. My comments are:The manuscript by Dr. Yanez Guera et al., describes the discovery of a novel sNPF-like signaling system in echinoderms and uses this information to devise a scheme for the evolution of the NPY/NPF versus sNPF signaling in the bilateria. I found the paper to be difficult to read because it was a hybrid between a research report and a review. Also, it took a historical approach to presenting the data, which made it difficult to follow and to understand why certain topics were being covered.

We have removed non-essential sections of the Introduction and Discussion so that only essential background information is included. Also we have changed the structure of the paper away from an “historical approach to presenting the data” so that the logic of the paper is easier to follow.

One example is their focus in the beginning on the NPY/NPF system. This was obviously relevant when the authors first thought that their peptide was an echinoderm member of this family and they referred to it as ArNPYLP. However, neither the gene structure nor the peptide sequence supported placement in this family. A strong point to the paper is that to better understand their starfish peptide they sought its receptor. They could not find NPY/NPF-like receptors in starfish but they did find a receptor related to protostome sNPF receptor that did bind the peptide with sub-nanomolar affinity. At this point they then changed the name of the peptide to Ar-sNPF. I found this name switch in mid-paper to be quite confusing!

We agree that this was an unnecessarily confusing aspect of the original version of the paper. We have now avoided this confusion by highlighting at the outset of the paper (Figure 1) the similarity that the echinoderm peptides share with PrRPs and naming them accordingly (e.g. ArPrRP).

The main point of the paper is that the sNPF system is not confined to the protostomes as previously thought. However, I think that the data supporting that echinoderms have a sNPF signaling system is weak. They show that the gene structure of their Ar-sNPF has no resemblance to known sNPF genes from protostomes (Figure ). In comparing Ar-sNPF to other sNPF's the authors speak of "modest" similarity, but most of the similarity is in the C-terminal RFamide which is shared by many other peptide families. Indeed, as they then show, the better sequence match is with PrRP-like peptides of hemicordates and the PrRP-type peptides of chordates. They would be on firmer ground calling the starfish molecule a PrRP-like peptide. The reason for concluding that the starfish peptide is used in a sNPF-like system then rested with the receptor. Jekely (2013) concludes that the receptors for the PrRP and sNPF peptides cluster together. This differs from Figure 3 of this paper, and it is not clear to me which interpretation is better supported. In their Figure, the echinoderm receptors are an out group to the rest of the sNPF-type receptors. I do not know the level of confidence in placing them with the sNPF type receptors rather than being over with the hemicordates and the PrRP-type receptors. As it stands right now, the hemicordates and the echinoderms have similar peptides but they work through two different types of receptors.

We agree that the echinoderm peptides are more appropriately named as PrRP-type neuropeptides and we have changed the nomenclature accordingly (e.g. ArPrRP). Furthermore, by performing our own CLANS analysis of the receptor sequences we show that, consistent with the findings of Jekely (2013), PrRP-type receptors cluster with sNPF-type receptors. A novelty of our analysis is the inclusion of several echinoderm receptor sequences and, interestingly, the echinoderm receptors showed stronger connections with the sNPF/PrRP receptor cluster (Figure 3) than with the NPY/NPF receptor cluster. It is noteworthy, however, that whilst strong connections between the echinoderm receptors and PrRP/sNPF-type receptors in other taxa can be seen using CLANS, the lines linking to the echinoderm receptors are quite long (Figure 3). This suggests that the echinoderm receptors are orthologs of PrRP/sNPF-type receptors but have undergone sequence divergence. This divergence, which is also seen in nematode sNPF/PrRP-type receptors, probably explains why bilaterian sNPF/PrRP-type receptors do not form a monophyletic clade in phylogenetic tree-based analyses of receptor relationships (see new Figure 4). Our conclusion is that the sNPF-type and PrRP-type signalling systems are orthologous, with our discovery of a PrRP/sNPF-type signalling system in echinoderms providing key evidence in support of this, as summarised in a revised summary figure (Figure 6).

[Editors’ note: what follows is the authors’ response to the second round of review.]

Substantive Concerns:The inconsistency in receptor identification between the two methods and the long branches in the phylogenetic analysis are of concern.

Both methods (CLANS and phylogenetic tree) revealed a close relationship between the echinoderm receptors and protostome sNPF-type receptors, so in this respect there is not inconsistency. Furthermore, CLANS reveals that sNPF-type receptors are most closely related to PrRP-type receptors, whilst our discovery that a starfish PrRP-like peptide acts as a ligand for a sNPF/PrRP-type receptor provides experimental evidence that sNPF-type receptors and PrRP-type receptors are orthlogous. An alternative explanation would be that gene duplication in a common ancestor of the Bilateria gave rise to two sNPF/PrRP-type signalling systems, which were then differentially lost/retained in bilaterian lineages. However, in this non-parsimonious scenario gene loss would have to be invoked in multiple lineages, including protostomes, echinoderms, hemichordates and chordates. To show that we are aware of this alternative explanation, we added the following sentence to the Discussion:

“One possible explanation for this inconsistency would be that gene duplication in a common ancestor of the Bilateria gave rise to two sNPF/PrRP-type signalling systems, which were then differentially lost/retained in bilaterian lineages, but in such a scenario gene loss in several lineages would have to be invoked.”

We wonder if the authors have looked at the intron-exon structure of the receptors?

Yes we did investigate the intron-exon structure of the receptor genes but, unlike the precursor genes (Figure 2), this was unhelpful for investigation of orthology. This is because there are no introns in the protein-coding region of the echinoderm receptor genes and this is also a feature of sNPF-type, PrRP-type and NPY/NPF-type receptor genes in the majority of other bilaterian taxa.

The paper does not require and more experiments or analysis but there are a couple of points that the authors might address.1) In the Discussion, the section "Discovery of a PrRP/sNPF-type neuropeptide signaling system in echinoderms" is long and simply repeats the data that are presented in the Results. It is not really Discussion material and it could be dropped or greatly condensed without affecting the paper.

We agree that this section of the Discussion was too long and have removed text that describes specific details of the results, which were not necessary for this section of the paper.

2) In comparing the echinoderm peptides to PrRP-type or NPY/NPF type peptides, the authors point out that there are "thirteen other residues in the echinoderm peptides that are identical to equivalently positioned residues in at least one of the chordate or hemicordate PrRPs, as highlighted by the asterisks" In part B of Figure 1, using the same criteria for comparing the starfish peptide to known NPY/NPF type peptides, almost every residue is denoted by an asterisk. It is important to point out highly conserved residues, but the significance of the co-occurrence of a particular residue in a single other species is questionable. We suggest dropping that part of the analysis.

We agree that the comparisons highlighted with asterisks in Figure 1A and 1B are not very informative with respect to neuropeptide relatedness and therefore we have removed the asterisks and changed the text accordingly.